# INFOTOK: ADAPTIVE DISCRETE VIDEO TOKENIZER VIA INFORMATION-THEORETIC COMPRESSION

**Haotian Ye**[12†*]      **Qiyuan He**[3†]      **Jiaqi Han**[2]      **Puheng Li**[2]      **Jiaojiao Fan**[1]

**Zekun Hao**[1]      **Fitsum Reda**[1]      **Yogesh Balaji**[1]      **Huayu Chen**[2]      **Sheng Liu**[2]

**Angela Yao**[3]      **James Zou**[2]      **Stefano Ermon**[2]      **Haoxiang Wang**[1]      **Ming-Yu Liu**[1]

[1]NVIDIA      [2]Stanford University      [3]National University of Singapore

https://research.nvidia.com/labs/dir/infotok/

## ABSTRACT

Accurate and efficient discrete video tokenization is essential for long video sequences processing. Yet, the inherent complexity and variable information density of videos present a significant bottleneck for current tokenizers, which rigidly compress all content at a fixed rate, leading to redundancy or information loss. Drawing inspiration from Shannon's information theory, this paper introduces INFOTOK, a principled framework for adaptive video tokenization. We rigorously prove that existing data-agnostic training methods are suboptimal in representation length, and present a novel evidence lower bound (ELBO)-based algorithm that approaches theoretical optimality. Leveraging this framework, we develop a transformer-based adaptive compressor that enables adaptive tokenization. Empirical results demonstrate state-of-the-art compression performance, saving 20% tokens without influence on performance, and achieving $2.3\times$ compression rates while still outperforming prior heuristic adaptive approaches. By allocating tokens according to informational richness, INFOTOK enables a more compressed yet accurate tokenization for video representation, offering valuable insights for future research.

## 1 INTRODUCTION

With the growing success of vision foundation models (Agarwal et al., 2025; Zhang et al., 2025), there has been an increasing interest to model the real world with visual observations, e.g., videos. Representing video as discrete tokens offers clear advantages by aligning seamlessly with LLMs and unifying diverse vision tasks, yet discrete video tokenization remains highly challenging, especially for complex visual content. The main difficulty lies in developing efficient and scalable representations for long sequences, which can easily expand to millions of tokens and become intractable for the transformer architecture (Esser et al., 2021; Wang et al., 2024; Agarwal et al., 2025).

A discrete tokenizer typically consists of an encoder, a quantizer, and a decoder. Given a video with a certain duration and resolution, the encoder compresses the original video into a sequence of embeddings, which are subsequently quantized into discrete tokens. During decoding, the decoder reconstructs the video from these token sequences. The training objective aims to minimize the discrepancy between the original and reconstructed videos—an objective achievable if token sequences are allowed to be arbitrarily long (e.g., directly preserving all RGB pixel values). Hence, an effective tokenizer inherently acts as a compressor, encoding visual signals into compact tokens while preserving essential information for downstream tasks. Unlike images (Van Den Oord et al., 2017; Mentzer et al., 2023b; Tian et al., 2024; Yu et al., 2024), video signals pose unique tokenization challenges due to their high dimensionality, temporal redundancy, and considerable variability in

---

*Work done as an intern at NVIDIA and accepted as an Oral presentation at ICLR 2026. Code is available at here. Correspond to: haotianye@stanford.edu. † stands for equal contribution.

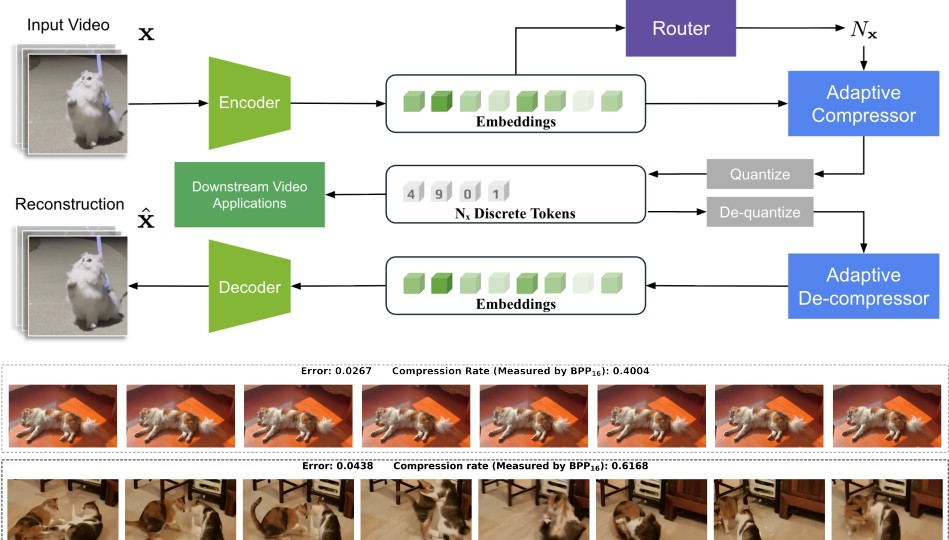

Figure 1: Overall framework of INFOTOK, an information-theoretic adaptive video tokenizer. An encoder maps video $\mathbf{x}$ into fixed-length embeddings, from which a router estimates the number of tokens $N_{\mathbf{x}}$ based on information complexity (section 3.1). An adaptive compressor encodes the embeddings to $N_{\mathbf{x}}$ tokens (section 3.2). For reconstruction, the tokens are decompressed to fixed-length embeddings and decoded back into video. INFOTOK tokenizes based on video complexity: e.g., the stable dog video is compressed more (0.40) than the dynamic cat-fighting video (0.62). Illustration details can be found in Appendix A.

information content (Yan et al., 2024; Agarwal et al., 2025). Therefore, developing an efficient, scalable, and principled video tokenizer is crucial for advancing unified multi-modal models.

As is well established in Shannon information theory (Ash, 2012), compared with content-agnostic compression that compresses all data identically, content-dependent compression, which determines the compression ratio of a data sample according to its property, is significantly more efficient. However, regardless of architectures and quantization methods, most existing tokenizers use an identical compression rate for all possible videos (Yan et al., 2021; Esser et al., 2021; Yu et al., 2023a). Consequently, a fixed amount of tokens could be redundant for simple videos and insufficient for representing complex videos, making downstream video-understanding or generation tasks inefficient or intractable. While recent attempts on flexible tokenization (Yan et al., 2024) use a heuristic training method that enables users to tokenize a video with varying lengths, we prove in this paper that their training method is biased, and their inference via trial-and-error length selection is inefficient. As a result, we seek to answer the following principled and challenging problem:

*What is a theoretically ideal discrete video tokenizer, and how to train this tokenizer in principle?*

To answer the question, this paper draws inspiration from Shannon's Source Coding Theorem (Shannon, 1948) and proves that existing tokenizers, either with a fixed compression rate or data-agnostic adaptive rates, are biased in the sense that the expected token length is significantly larger than the optimal length in order to achieve identical reconstruction quality. In contrast, tokenizing each video in a difficulty-aware manner, i.e., which is closely tied to the frequency of contents based on information theory, can guarantee a *near-optimal* compression rate in theory. Altogether, our theoretical results suggest a principled way for adaptive tokenization.

Inspired by the theorems, we propose a novel information-theoretic adaptive tokenization framework, named INFOTOK, that converts an existing fixed-compression tokenizer into an adaptive counterpart. As illustrated in fig. 1, it leverages a router to determine the token sequence lengths based on the *evidence lower-bound* (ELBO) of the negative log-likelihood of input videos, and a transformer-based adaptive compressor to compress fixed-length embeddings into a discrete token sequence with length assigned by the router. By bridging to information theory, INFOTOK can produce adaptive token sequences with sequence length conditioned on the information complexity of each frame, thereby reducing redundancy and ensuring each token represents a similar amount of information.

Extensive experimental results on video reconstruction datasets demonstrate the superiority of INFOTOK. Empirically, our comprehensive experiments show that INFOTOK can save approximately $50\%$ tokens without loss of reconstruction quality compared to state-of-the-art fixed-length tokenizers, and it can outperform previous adaptive tokenizers by an average compression rate of $2.3\times$ and number of evaluations (NFEs) by $11\times$. We also conduct ablation studies to show that INFOTOK aligns with our theoretical intuition well, and present illustrative examples for presentation.

The main contributions of this paper are summarized below.

- We prove rigorously based on Shannon's information theory that existing tokenizers with fixed or data-agnostic adaptive compression rates are inherently biased and inefficient.
- We propose INFOTOK, an information-theoretic adaptive tokenization framework that leverages an ELBO-based router to dynamically decide compression rate, and a transformer-based adaptive compressor to compress fixed-length embeddings into adaptive token sequences effectively.
- Extensive empirical studies validate the superiority of INFOTOK, demonstrating significantly improved token efficiency without compromising quality.

## 2 ADAPTIVE TOKENIZATION: THE FRAMEWORK

### 2.1 DISCRETE VIDEO TOKENIZATION

We consider video data[1] represented as $\mathbf{x} \in \mathcal{X} \triangleq \{0, \cdots, 255\}^{T \times H \times W \times 3}$ where $T$ is the number of frames, and $H, W$ is the resolution. A discrete video tokenizer $\mathcal{T}$ parameterized by $\phi, \theta$ consists of: 1) an *encoder* $\mathcal{E}_\phi : \mathcal{X} \mapsto \mathbb{R}^{N \times K}$ that maps the input video into a sequence of continuous latents $\mathbf{h} = \mathcal{E}_\phi(\mathbf{x})$ with latent dimension $K$; 2) a *quantizer* $\mathcal{Q} : \mathbb{R}^K \mapsto \mathcal{Z}$, e.g., VQ (Gray, 1984), FSQ (Mentzer et al., 2023a), or LFQ (Yu et al., 2023b), that quantizes embeddings $\mathbf{h}$ into a sequence of discrete tokens $\mathbf{z} = \mathcal{Q}(\mathbf{h})$ (and de-quantizes $\mathbf{z}$ back to $\hat{\mathbf{h}} = \mathcal{Q}^{-1}(\mathbf{z})$), where $\mathcal{Z}$ is the codebook with size $C$; 3) a *decoder* $\mathcal{D}_\theta : \mathbb{R}^{N \times K} \mapsto \mathcal{X}$ that decodes from $\hat{\mathbf{h}}$ back to the video space through $\hat{\mathbf{x}} = \mathcal{D}_\theta(\hat{\mathbf{h}})$. Here, $N$ is the length of the token sequence, and since lengths and resolutions vary from video to video, $N$ is typically represented by $c \cdot THW$, where $c$ is a pre-defined compression rate. We refer to this as fixed-compression or fixed-length tokenizers, as when $T, H, W$ is given, the number of tokens is fixed. The tokenizer is optimized by minimizing the reconstruction loss:

$$\mathcal{L}_{\text{recon}}(\mathcal{T}) = \mathbb{E}_{\mathbf{x} \sim \mathbb{D}, q_\phi(\mathbf{z}|\mathbf{x})} \left[ -\log p_\theta(\mathbf{x}|\mathbf{z}) \right], \tag{1}$$

where $\mathbb{D}$ is the video distribution to sample $\mathbf{x}$ from, and $q_\phi(\mathbf{z}|\mathbf{x})$ and $p_\theta(\mathbf{x}|\mathbf{z})$ are the posteriors for compression and reconstruction parameterized by the encoder $\mathcal{E}_\phi$ and decoder $\mathcal{D}_\theta$, respectively. Under the standard assumption of VAE (Kingma et al., 2019), eq. (1) can be converted into a MSE loss on the pixel space. As such, tokenizers are essentially compressors that compress $\mathbf{x}$ into a dense discrete sequence $\mathbf{z}$.

### 2.2 A UNIFIED FRAMEWORK FOR ADAPTIVE TOKENIZATION

In this section, we analyze the inherent limitations of fixed-compression discrete tokenizers. We show that such a constraint is sub-optimal due to the varying spatio-temporal complexity across videos. Specifically, different videos contain diverse spatial content (e.g., scenes, objects) and exhibit varying temporal dynamics (e.g., speed, magnitude of motion), resulting in unequal informational demands. From the perspective of information theory, frequent contents are naturally easier to represent than infrequent contents, leading to a natural result of varying representation lengths. To better illustrate this sub-optimality, consider an idealized scenario where the tokenizer $\mathcal{T}$ can perfectly reconstruct any input video $\mathbf{x}$ with non-zero probability $p(\mathbf{x}) > 0$. In what follows, we restate the Shannon Source Coding Theorem (Shannon et al., 1959) from Information Theory to ground our argument.

**Theorem 2.1** (Shannon Source Coding Theorem (restated) (Shannon et al., 1959))**.** *For any tokenizer* $\mathcal{T}$ *with codebook size* $C$ *that can fully reconstruct video data* $\mathbf{x} \sim p(\mathbf{x})$ *defined above, we have*

$$\mathbb{E}_{\mathbf{x} \sim p(\mathbf{x})}[N_\mathbf{x}] \geq H_C(\mathbb{D}) \triangleq \mathbb{E}_{\mathbf{x} \sim p(\mathbf{x})}[-\log_C p(\mathbf{x})],$$

*where* $N_\mathbf{x}$ *is the token sequence length of* $\mathbf{x}$ *assigned by* $\mathcal{T}$. *Additionally, there exists an adaptive tokenization that has*

$$H_C(\mathbb{D}) \leq \mathbb{E}_{\mathbf{x} \sim p(\mathbf{x})}[N_\mathbf{x}] < H_C(\mathbb{D}) + 1.$$

---

[1]This paper considers the standard RGB pixel space, but our results can be naturally extended beyond. While we focus on video, the entire framework applies directly to images. We do not study image tokenization since the amount of tokens used for image representation is relatively small, and the incentive to optimize it is insufficient.

In contrast to theorem 2.1, it is obvious that any fixed-length tokenizer requires at least $\log_C(|\mathbb{D}|)$ tokens to fully reconstruct the distribution, which is significantly larger than $H_C(\mathbb{D})$ when the data distribution is not a uniform distribution. While this is a simplified case, it motivates us to design a framework for *adaptive* video tokenizer, which is presented below.

**Adaptive video tokenization.** Given a fixed-length tokenizer $\mathcal{T}$, an adaptive video tokenizer $\mathcal{T}_{\text{adaptive}} = (\mathcal{T}, r, \mathcal{M}_\psi)$ consists of the two extra key components: A router $r(N_{\mathbf{x}}|\mathbf{x})$ (could be deterministic or stochastic) that determines the token length $N_{\mathbf{x}}$ for each video $\mathbf{x}$, and an adaptive compressor $\mathcal{M}_\psi : \mathbb{R}^{N \times K} \times \mathbb{N}^+ \mapsto \mathbb{R}^{N_{\mathbf{x}} \times K}$ that effectively compresses the information into $N_{\mathbf{x}}$ tokens for more condensed representation. Before fine-grained analysis on each component, we present a pseudo-code of the framework in algorithm 1.

The role of the router is to determine an appropriate token length $N_{\mathbf{x}}$, which is the most important problem in adaptive tokenization. The adaptive compressor feeds the latent sequence $\mathbf{h}$ into a network that transforms information appropriately and outputs $\mathbf{h}'$ with length $N_{\mathbf{x}}$. The sequence $\mathbf{h}'$ is passed through a quantizer to obtain the discrete token sequence $\mathbf{z} = \mathcal{Q}(\mathbf{h}')$. For the decoding counterpart, it decompresses using $\mathcal{M}_\psi^{-1}$ that conversely transforms the dequantized sequence $\hat{\mathbf{h}}' = \mathcal{Q}^{-1}(\mathbf{z})$ with length $N_{\mathbf{x}}$ back to $\hat{\mathbf{h}}$ with the original length $N$. Finally, the reconstructed video is obtained via the decoder $\hat{\mathbf{x}} = \mathcal{D}_\theta(\hat{\mathbf{h}})$.

---

**Algorithm 1** Adaptive Tokenizer Training

**Input:** Encoder $\mathcal{E}_\phi$, decoder $\mathcal{D}_\theta$, router $r(N|\mathbf{x})$, adaptive compressor $\mathcal{M}_\psi$, video data distribution $\mathbb{D}$.

1: **repeat**
2:      Sample video $\mathbf{x} \sim \mathbb{D}$
3:      $\mathbf{h} \leftarrow \mathcal{E}_\phi(\mathbf{x})$                    ▷ Encode
4:      $N_{\mathbf{x}} \sim r(N|\mathbf{x})$              ▷ Router § 3.1
5:      $\mathbf{h}' \leftarrow \mathcal{M}_\psi(\mathbf{h}, N_{\mathbf{x}})$      ▷ Compressor § 3.2
6:      $\mathbf{z} \leftarrow \mathcal{Q}(\mathbf{h}')$                   ▷ Quantize
7:      $\hat{\mathbf{h}}' \leftarrow \mathcal{Q}^{-1}(\mathbf{z})$                  ▷ Dequantize
8:      $\hat{\mathbf{h}} \leftarrow \mathcal{M}_\psi^{-1}(\hat{\mathbf{h}}', N_{\mathbf{x}})$   ▷ Decompressor § 3.2
9:      $\hat{\mathbf{x}} \leftarrow \mathcal{D}_\theta(\hat{\mathbf{h}})$                    ▷ Decode
10:     Optimize $\mathcal{L}_{\text{recon}}^{\text{adaptive}}$ per Eq. 4
11: **until** converged

---

Similarly, the adaptive tokenizer is optimized by the reconstruction loss

$$\mathcal{L}_{\text{recon}}(\mathcal{T}_{\text{adaptive}}) = \mathbb{E}_{\mathbf{x} \sim \mathbb{D}, N_{\mathbf{x}} \sim r(N_{\mathbf{x}}|\mathbf{x}), \mathbf{z} \sim q_{\phi, \psi}(\mathbf{z}|\mathbf{x}, N_{\mathbf{x}})} \left[ -\log p_{\theta, \psi}(\mathbf{x}|\mathbf{z}, N_{\mathbf{x}}) \right], \qquad (2)$$

which is depicted in Algorithm 1. Remarkably, we design our adaptive tokenizer framework on top of existing fixed-length tokenizers, ensuring seamless compatibility and enabling it to benefit directly from future advancements in the field. In short, *the objective of adaptive tokenization is to leverage its additional flexibility to compress token sequences in a principled way without sacrificing performance, thereby representing long video more accurately and efficiently.*

## 2.3 SUB-OPTIMALITY OF EXISTING METHODS

Specifying $N_{\mathbf{x}}$ for each video is challenging because adaptiveness is not only for flexibility but also for principled compression. As special cases of our framework, several existing tokenizers for images (Duggal et al., 2024) and videos (Duggal et al., 2024) propose to train with a heuristic and data-agnostic router, where $N_{\mathbf{x}}$ is sampled *uniformly* from $\{1, \cdots, N\}$. During inference, the sequence length is selected based on the quality. However, uniform selection does not take into account the different information quantities possessed by different videos. Furthermore, the uncertainty in random selection brings additional overhead at inference time: To determine the optimal length, one needs to perform a search over possible lengths. To rigorously prove why this is biased, we simplify their inference stage and assume that an oracle $r^*(N_{\mathbf{x}}|\mathbf{x})$ can directly return the minimal sequence length conditioned on the loss eq. (2) being minimized. The complete proof is deferred to section B.

**Theorem 2.2.** *Assume that $r(\cdot|\mathbf{x})$ is a uniform distribution over $\{1, \cdots, N\}$. For any constant $\kappa > 1$, there exists a data distribution $\mathbb{D}$ and a sufficiently large $N$, such that for any adaptive tokenizer $\mathcal{T}_{\text{adaptive}}^* = (\mathcal{T}^*, r, \mathcal{M}_\psi^*)$ that minimizes eq. (2), any $r^*(\cdot|\mathbf{x}) \in \arg\min_r \mathcal{L}_{recon}((\mathcal{T}^*, r, \mathcal{M}_\psi^*))$,*

$$\mathbb{E}_{\mathbf{x} \sim p(\mathbf{x}), N_{\mathbf{x}} \sim r^*(N_{\mathbf{x}}|\mathbf{x})}[N_{\mathbf{x}}] \geq \kappa H_C(\mathbb{D}).$$

**Intuition.** We briefly describe the underlying intuition of the theorem. While a uniform $r$ requires the model to reconstruct videos simultaneously at different information levels, it does not inject incentives for reducing the expected token length. Consequently, data with different likelihoods are treated indifferently in this loss function, and the tokenizer could be trained to reduce the prediction difficulty when the sampled $N_{\mathbf{x}}$ is small, even though it has a negative influence to the average token length. As an illustrative example, readers can consider the tokenization problem of a four-data distribution

with probability $\{2^{-1}, 2^{-2}, 2^{-3}, 2^{-3}\}$, with codebook size $C = 2$. While the optimality is to use $1, 2, 3, 3$ tokens for each video, minimizing the above loss results in all videos consuming 2 tokens, an inefficiency that will exacerbate as the distribution becomes more complex and imbalanced.

Theorem 2.2 highlights the intrinsic bias of the data-agnostic router: even if we manage to minimize the loss during training, the uniform pattern of the router will bias the tokenizer to tokenize data in a way that, during inference, *its expected token sequence length could be arbitrarily large compared to the optimality*. Remarkably, while this theorem is particular for the uniform router and the case when the loss is minimized, it demonstrates the fundamental limitation of data-agnostic router for training.

# 3 INFOTOK: INFORMATION-THEORETIC ADAPTIVE TOKENIZATION

Noticing the limitations of the data-agnostic routers, an intuitive alternative is to add the expected sequence length to the loss function, as the goal is to minimize it while maintaining good reconstruction quality. Unfortunately, $N_\mathbf{x}$ is discrete and not directly optimizable, making this purely optimization-based solution intractable. This section proposes a novel router that aligns with the information theory, thus circumventing the dilemma of directly optimizing sequence length.

## 3.1 INFORMATION-THEORETIC ADAPTIVE TOKENIZATION

**Token length selection via ELBO.** Our theory in section 2.2 has indicated that, in order to achieve an optimal expected token length, $N_\mathbf{x}$ should be proportional to the negative log-likelihood. In other words, the minimized $H_C(\mathbb{D})$ is realized by setting $N_\mathbf{x} = -\log p(\mathbf{x})$. Yet, the log-likelihood of arbitrary video $\mathbf{x}$ is intractable and thus estimation is required. We propose to leverage the Evidence Lower Bound (ELBO) as a surrogate, which is defined as:

$$\text{ELBO}(\mathbf{x}) = \mathbb{E}_{q_\phi(\mathbf{z}|\mathbf{x})}\left[\log p_\theta(\mathbf{x}|\mathbf{z}) - D_{\text{KL}}\left[q_\phi(\mathbf{z}|\mathbf{x})\|p(\mathbf{z})\right]\right], \tag{3}$$

where $q_\phi(\mathbf{z}|\mathbf{x})$ and $p_\theta(\mathbf{x}|\mathbf{z})$ are the encoding and decoding distributions, parameterized by the encoder $\mathcal{E}_\phi$ and decoder $\mathcal{D}_\theta$ respectively, and $p(\mathbf{z})$ is the prior. Our choice to employ ELBO is motivated as ELBO is a provable lower bound on the log-likelihood of data, *i.e.*, $\text{ELBO}(\mathbf{x}) \leq \log p(\mathbf{x})$. Critically, the model is trained directly to minimize the gap, and the bound becomes tight when the approximate posterior approaches the true posterior, *i.e.*, $D_{\text{KL}}\left[q_\phi(\mathbf{z}|\mathbf{x})\|p_\theta(\mathbf{z}|\mathbf{x})\right] \approx 0$, making ELBO a reasonable approximation to the log-likelihood.

Specifically, we design the router for token length selection as

$$r_\beta(N_\mathbf{x}|\mathbf{x}) = \delta(\beta \cdot \frac{\text{ELBO}(\mathbf{x})}{\mathbb{E}[\text{ELBO}(\mathbf{x})]}), \tag{4}$$

where $\beta$ is the average compression factor that measures how much information a token should represent, $\delta(\cdot)$ denotes the delta distribution, and the term $\mathbb{E}[\text{ELBO}(\mathbf{x})]$ is for normalization. In addition to the theoretical merit that is discussed below, our choice directly approximates the optimal sequence length through eq. (4) and thus does not require extensive probing over different lengths during both training and inference as Yan et al. (2024). In practice, eq. (4) can be computed efficiently: we first encode $\mathbf{x}$ and decode back to $\hat{\mathbf{x}}$ without using the adaptive compressor, so as to compute the reconstruction error, which is the negative ELBO when adding the additional KL term. We then reuse the continuous embedding $\mathbf{h}$ and then compute $\mathbf{h}' = \mathcal{M}_\psi(\mathbf{h}, N_\mathbf{x})$, which is henceforth quantized into a token sequence $\mathbf{z}$ with length $N_\mathbf{x}$. As a result, we only have one additional decoder pass in order to generate the compressed token sequence.

Unsurprisingly, our method can achieve near-optimal token allocation, as stated below.

**Theorem 3.1.** *For any adaptive tokenizer $\mathcal{T}_{adaptive} = (\mathcal{T}, r, \mathcal{M}_\psi)$ with the router $r$ defined as eq. (4) and $\beta \geq -\mathbb{E}[\text{ELBO}(\mathbf{x})]$, if the tokenizer manages to minimize the reconstruction loss, i.e.,*

$$\mathcal{T}_{adaptive}^* \in \arg\min_{\mathcal{T}_{adaptive}} \mathcal{L}_{\text{recon}}(\mathcal{T}_{adaptive}),$$

*then during inference, $\mathcal{T}_{adaptive}^*$ guarantees that*

$$\mathbb{E}_{\mathbf{x}\sim p(\mathbf{x}), N_\mathbf{x}\sim r(N_\mathbf{x}|\mathbf{x})}[N_\mathbf{x}] \leq H_C(\mathbb{D}) + \beta - \mathbb{E}_{\mathbf{x}\sim p(\mathbf{x})}[-\log p(\mathbf{x})].$$

The proof is deferred to section B. Theorem 3.1 implies that if the tokenizer is well-trained to minimize the loss, then the compression rate of INFOTOK is optimal up to the approximation error.

Notice that $\beta \geq -\mathbb{E}[\text{ELBO}(\mathbf{x})] \geq \mathbb{E}[-\log p(\mathbf{x})]$ by the definition of ELBO. Provided by large-scale neural networks (fixed-length tokenizers), ELBO values are believed to be close enough to the log-likelihoods, thus providing a strong theoretical guarantee when $\beta$ is selected appropriately.

In practice, $\beta$ can be chosen up-front based on the computation budget to determine the average amount of tokens used. To further simplify the problems, we can ensemble multiple choices of $\beta$ into one tokenizer, i.e., we allow $\beta$ to be chosen from different values and input $\beta$ to the adaptive compressor during training. As such, this single compressor can understand the amount of information to put in each token with different $\beta$, thus remaining performant under a diverse range of compression rates. We name this approach INFOTOK-Flex with $r_\beta^{\text{flex}}(N_\mathbf{x}|\mathbf{x}) = \frac{1}{\mathcal{B}}\sum_{\beta \in \mathcal{B}} r_\beta(N_\mathbf{x}|\mathbf{x})$ for some given $\mathcal{B}$. Empirically, we find that using the reconstruction error itself (without the KL term) to derive $r_\beta(N_\mathbf{x}|\mathbf{x})$ is sufficient, as the KL term is approximately proportional to the reconstruction error, and the ratio is similar. During inference, INFOTOK-Flex automatically tokenizes each video into a token sequence based on the given $\beta$, achieving adaptive tokenization.

## 3.2 ADAPTIVE COMPRESSOR

The remaining question is how to compress $\mathbf{h}$ into a shorter $\mathbf{h}'$ via the adaptive compressor, given a token budget $N_\mathbf{x}$. For ideal fixed-length tokenizers where tokens are organized as a 1D sequence without geometric bias, merging information into the first $N_\mathbf{x}$ tokens and discarding the remaining $N_{\max} - N_\mathbf{x}$ tokens could be a solution. However, existing SOTA tokenizers that support multiple resolutions and video lengths are spatial-temporal sensitive, making the above approach less reasonable (verified in our experiments). In addition, the high-dimensionality of video data naturally poses extra space to consider, *e.g.*, the variability within each video.

**Likelihood-based token selection.** To address this issue, we introduce a design for the adaptive compressor that takes into account the variability of information content within a video by discarding the tokens with the *lowest information content*. Aligned with our analysis in section 3, we preserve the top $N_\mathbf{x}$ tokens according to their corresponding per-token log-likelihood, which is also approximated via the ELBO values. Specifically, the adaptive compressor computes a binary mask $\mathbf{m} \in \{0, 1\}^N$ where $N_\mathbf{x}$ tokens with the lowest ELBO values are $1$ and the remaining are $0$. Interestingly, it does not incur extra network evaluation since the log-likelihood term has been computed in the router $r(\cdot|\mathbf{x})$. The end-to-end reconstruction loss will train the adaptive compressor to transform information in tokens to be masked to the remaining positions. To ensure that the video can be decoded at inference time, $\mathbf{m}$ is stored as part of the discrete token sequence $\mathbf{z}$, leading to a minimal overhead of approximately $5\%$ in token length, which is worthwhile as we will see in experiments.

**Architecture.** As a general framework, INFOTOK can be seamlessly integrated on top of established tokenizer architectures by reusing their encoder $\mathcal{E}_\phi$ and decoder $\mathcal{D}_\theta$. Empirically, we implement INFO-TOK based on Cosmos Discrete Video Tokenizer (Agarwal et al., 2025), a 3D-CNN-based fixed-length tokenizer used to initialize $\mathcal{E}_\phi$ and $\mathcal{D}_\theta$. To account for the extra flexibility introduced in the adaptive compressor and decompressor, we additionally initialize the compressor $\mathcal{M}_\psi$ with a stack of multiple Transformer layers to effectively compress $\mathbf{h}$ into $\mathbf{h}'$. Similarly, the decompressor contains multiple Transformer layers for mapping the condensed representation back to the full-length sequence.

## 4 EXPERIMENT

We present experimental results about the superiority of INFOTOK in video tokenization performance in this section. Notice that this paper focuses mainly on adaptive *compression* that represents videos accurately with fewer tokens. While video *generation* is an important downstream application, training a video generative model is extremely resource-consuming and is beyond our scope. For more details about training, inference, and resource consumption, please refer to Appendix C.

## 4.1 EXPERIMENTAL SETTINGS

**Dataset.** To systematically evaluate different tokenization, we consider two video reconstruction datasets, namely TokenBench proposed in Agarwal et al. (2025), and DAVIS in Caelles et al. (2019), each containing videos with various resolutions and numbers of frames. Since ElasticTok only takes in square 256px video with $H = W = 256$, we only take the 256px partition of both datasets and crop each video to a square shape randomly. All results below are reported on our processed datasets for fair comparisons, although this will make baseline results not directly comparable with their paper. Notably, InfoTok can generalize to video with different resolutions, as presented in Appendix. D.

Table 1: Evaluation of fixed-length and adaptive tokenizers on TokenBench and DAVIS. We compare INFOTOK with ElasticTok at two compression levels (0.81, 0.56) by setting our compression rates to theirs.

| Tokenization Method | Compression (BPP$_{16}$ ↓) | TokenBench-256x256 | | | | DAVIS-256x256 | | | |
|---|---|---|---|---|---|---|---|---|---|
| | | PSNR↑ | SSIM↑ | LPIPS↓ | FVD↓ | PSNR↑ | SSIM↑ | LPIPS↓ | FVD↓ |
| Open-MAGVIT2-UCF | 1.12 | 25.77 | 0.762 | 0.269 | 121 | 21.98 | 0.656 | 0.317 | 583 |
| OmniTokenizer | 0.81 | 23.15 | 0.767 | 0.276 | 152 | 20.05 | 0.699 | 0.318 | 673 |
| Cosmos-DV4x8x8 | 1.00 | 30.01 | 0.885 | 0.138 | 49 | 25.92 | 0.793 | 0.208 | 404 |
| ElasticTok | 0.81 | 28.26 | 0.84 | 0.244 | 141 | 24.69 | 0.752 | 0.318 | 754 |
| INFOTOK-Flex | 0.81 | 29.86 | 0.878 | 0.148 | 54 | 25.69 | 0.781 | **0.216** | 441 |
| INFOTOK | 0.81 | **30.08** | **0.881** | **0.145** | **49** | **25.79** | **0.788** | 0.223 | **408** |
| ElasticTok | 0.56 | 27.34 | 0.813 | 0.276 | 194 | 23.76 | 0.714 | 0.356 | 930 |
| INFOTOK-Flex | 0.56 | **29.30** | **0.857** | 0.179 | 71 | **24.84** | **0.742** | **0.274** | 581 |
| INFOTOK | 0.56 | 29.27 | 0.854 | **0.176** | **70** | 24.52 | 0.738 | 0.277 | **540** |

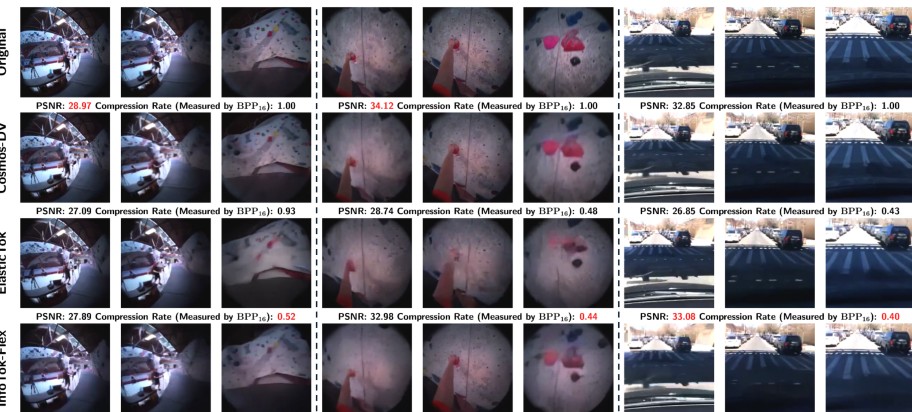

Figure 2: Reconstructions examples of video with different complexities using different tokenizers. INFO-TOK-Flex can achieve similar PSNR with much higher compression (compared to Cosmos-DV), and similar compression rates with better PSNR (compared to ElasticTok).

**Evaluations.** We evaluate the quality of reconstructed videos by four metrics: the Peak Signal to Noise Ratio (PSNR) (Hore & Ziou, 2010), the Structural Similarity (SSIM) (Hore & Ziou, 2010), the Learned Perceptual Image Patch Similarity (LPIPS) metric (Zhang et al., 2018), and the Fréchet Video Distance (Unterthiner et al., 2019). To evaluate the compression, we adopt the bits-per-pixel (BPP) value that computes how many digital bits are used to represent one pixel (with all color channels) in the whole video. For the sake of presentation, we present BPP with a unit of $\frac{1}{16}$, which is represented as BPP$_{16}$ (bits-per-16-pixels). For instance, if the number of tokens is $c \cdot THW$ for a video with resolution $H \times W$ and length $T$, and the codebook size is $C$, then BPP$_{16} = 16c \cdot \log(C)$.

**INFOTOK Details.** We use Cosmos Tokenizer (Agarwal et al., 2025) as the fixed-length tokenizer and use it to compute the ELBO value of inputs. Given a video of shape $T \times H \times W$, it uses $N_{max} = \frac{T}{4} \times \frac{H}{8} \times \frac{W}{8} = \frac{1}{256}THW$ tokens to tokenize the video. We refer our methods using the router $r_\beta(N_\mathbf{x}|\mathbf{x})$ as INFOTOK and the one using $r_\beta^{flex}(N_\mathbf{x}|\mathbf{x})$ as INFOTOK-Flex. For router $r_\beta(N_\mathbf{x}|\mathbf{x})$, $\mathbb{E}[\text{ELBO}(\mathbf{x})]$ is computed from the exponential moving average of historical training samples. For router $r_\beta^{flex}(N_\mathbf{x}|\mathbf{x})$, we consider $\mathcal{B} = \{0.25N_{max}, 0.5N_{max}, 0.75N_{max}, N_{max}\}$ to assemble tokenizers with different average compression rates. During inference, we specify an average BPP$_{16}$, and $\beta$ can be computed as $N_{max} \cdot (\text{BPP}_{16} - \frac{1}{16})$, where $\frac{1}{16}$ is the cost of binary mask. The adaptive compressor is an eight-layer transformer with a block-causal attention matrix that maintains the causality of the Cosmos Tokenizer. For quantization, we use the Finite Scalar Quantizer (Mentzer et al., 2023b).

**Baselines.** First, for fixed-compression tokenizers, we evaluate Open-MAGVIT2 (codebook size $C = 2^{18}$) (Yu et al., 2023a), OmniTokenizer ($C = 2^{13}$) (Wang et al., 2024), and the Cosmos Discrete Video Tokenizer ($C = 2^{16}$) (Agarwal et al., 2025). Their resulting BPP$_{16}$ values are reported in Table 1. For adaptive tokenization, we compare against ElasticTok ($2^{16}$) (Yan et al., 2024), which employs right-to-left random masking on token sequence during training and binary search to select the minimal number of tokens to meet a specified reconstruction-loss threshold, with BPP$_{16}$ ranging from 0.25 to 4. Since ElasticTok's adaptiveness is governed by a loss threshold rather than an average compression rate, we align our methods with their settings.

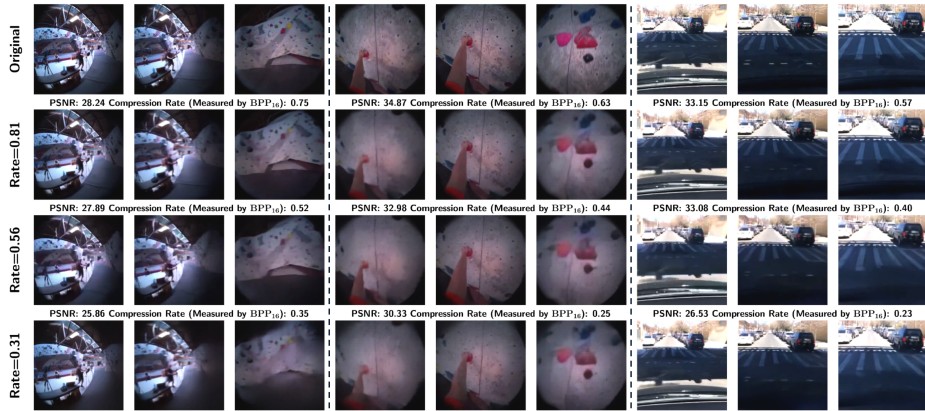

Figure 3: Reconstructions examples of video by INFOTOK-Flex with different compression rates.

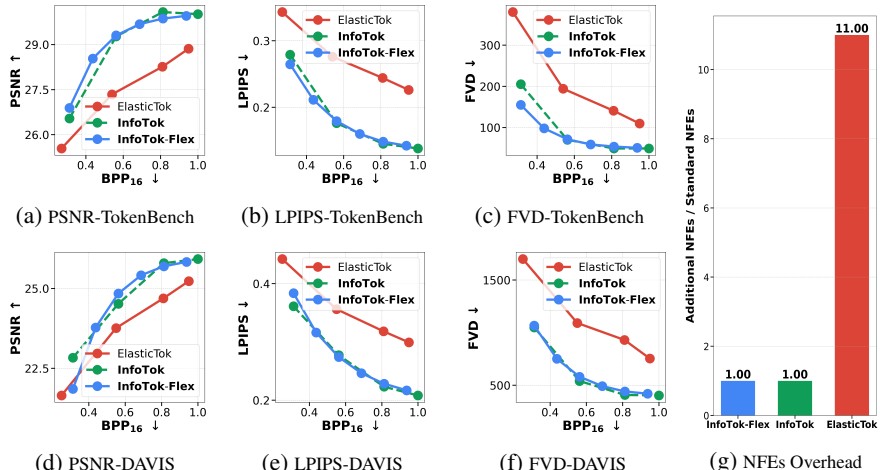

Figure 4: Video tokenization performance of INFOTOK-Flex, INFOTOK, and ElasticTok on TokenBench (a-c) and DAVIS (d-f). Quality metrics are plotted against $BPP_{16}$ (bits per 16 pixels). Tokenization efficiency measured in the Number of Function Evaluations overhead (additional NFEs / standard NFEs ↓) is shown in (g). InfoTok-Flex and InfoTok achieve superior reconstruction quality with smaller $BPP_{16}$ levels. Additionally, INFOTOK is significantly more efficient than ElasticTok, which requires searching to meet thresholds.

### 4.2 SUPERIORITY OF INFOTOK

**Benchmarking Results.** Table 1 compares our methods against several baselines at average compression rates of BPP16 = 0.81 and 0.56. Compared to ElasticTok, INFOTOK-Flex and INFOTOK achieve notably better reconstruction quality at the same average compression levels, with FVD reduced by $40 \sim 60\%$, LPIPS by $25 \sim 40\%$, and PSNR by $1.0 \sim 2.0$ (which is MSE reduced by $20\% \sim 36\%$). Remarkably, INFOTOK can outperform ElasticTok with $BPP_{16} = 0.81$ with a lower compression rate (0.56). Compared to fixed-compression baselines, INFOTOK performs similarly to Cosmos-DV with $20\%$ tokens saved and outperforms the remaining baselines even when the average $BPP_{16}$ is cut in half, again demonstrating the advantages of adaptive tokenization.

**Token Efficiency with Flexibility.** Figure 4 provides a more comprehensive comparison with ElasticTok under different average compression rates. The blue line represents the performance of INFOTOK-Flex trained with $r_\beta^{\text{flex}}$ and used at different compression rates $\beta$, and the green line represents INFOTOK trained on different compression rate levels and used at those levels. The red line represents the ElasticTok model with different loss thresholds, resulting in different performance and compression rates. Improvements over ElasticTok are salient, with similar reconstruction quality (PSNR) and perceptual scores (LPIPS, FVD) achieved by compression rates $2.3\times$ larger (e.g., PSNR on TokenBench, FVD on DAVIS). Notably, at the same inference setting, INFOTOK-Flex performs on par with INFOTOK trained with different specific $r_\beta$, implying that it can ensemble models with varying rates while still outperforming ElasticTok, which is trained with a uniform router.

Table 2: Ablation on INFOTOK versus an optimal search-based strategy to determine the token lengths. "Optimal" is a strict upper bound of our method, yet their performance is extremely close.

| Compression ($BPP_{16}$ ↓) | Inference Method | TokenBench-256x256 | | | | DAVIS-256x256 | | | |
|---|---|---|---|---|---|---|---|---|---|
| | | PSNR↑ | SSIM↑ | LPIPS↓ | FVD↓ | PSNR↑ | SSIM↑ | LPIPS↓ | FVD↓ |
| 0.81 | INFOTOK-Flex | 29.86 | 0.878 | 0.148 | 54 | 25.69 | 0.782 | 0.228 | 441 |
| 0.81 | Optimal | 29.92 | 0.878 | 0.148 | 54 | 25.79 | 0.784 | 0.226 | 431 |
| 0.56 | INFOTOK-Flex | 29.30 | 0.857 | 0.179 | 71 | 24.84 | 0.742 | 0.274 | 581 |
| 0.56 | Optimal | 29.39 | 0.856 | 0.180 | 74 | 24.93 | 0.740 | 0.276 | 601 |
| 0.31 | INFOTOK-Flex | 26.89 | 0.782 | 0.265 | 155 | 21.85 | 0.639 | 0.384 | 1066 |
| 0.31 | Optimal | 26.98 | 0.779 | 0.267 | 165 | 21.97 | 0.640 | 0.388 | 1138 |

**Inference Efficiency.** Figure 4g illustrates the advantage of our inference efficiency. To determine the token length for a video, ElasticTok runs a binary search over each 4096-token block sequentially to satisfy the given loss threshold, requiring an additional $\log_2(4096) - 1 = 11$ network forward evaluations (NFEs). In contrast, both INFOTOK and INFOTOK-Flex need only one additional network evaluation of the decoder to compute the ELBO, demonstrating substantial efficiency gains. More details regarding wall-clock inference latency comparison can be found in Appendix D.

## 4.3 ABLATION STUDY

**Ablation on routers.** To evaluate how well our ELBO-based token allocation performs, we compare it to an optimal routing strategy. For each video, it exhaustively evaluates the reconstruction quality with $BPP_{16} \in \left\{ \frac{1}{16}, \frac{6}{16}, \ldots, 1 \right\}$, and then solves the optimization problem over the entire dataset to find the optimal performance constrained on the average compression rate. As shown in Table 2, the ELBO-based router achieves similar performance compared with the optimal strategy, aligning well with our theory and demonstrating the feasibility to specify token lengths without brute-force search.

Table 3: Ablation results on TokenBench with an average $BPP_{16} = 0.56$. (Left) Ablation on adaptive compressors. (Right) Ablation on different variants of adaptive mechanisms across architectures.

| Compressor | PSNR | SSIM | LPIPS | FVD |
|---|---|---|---|---|
| R2L | 27.43 | 0.825 | 0.207 | 137 |
| Jump | 28.07 | 0.841 | 0.173 | 84 |
| Ours | 29.30 | 0.857 | 0.179 | 71 |

| Architecture | Adaptive Mechanism | PSNR | FVD |
|---|---|---|---|
| Cosmos | Uniform (ElasticTok) | 27.35 | 152 |
| | ELBO (InfoTok) | 29.30 | 71 |
| ElasticTok Backbone Vision Transformer | Uniform (ElasticTok) | 27.21 | 198 |
| | ELBO (InfoTok) | 28.64 | 114 |

**Ablation on adaptive compressor.** To better understand the ELBO-based compression, we compare it with two additional intuitive strategies: simple right-to-left masking used by ElasticTok (R2L), and a spatially dispersed variant that masks a token every four tokens (Jump), in Table 3 (Left). Our compressor that masks tokens by ELBO values yields better reconstruction quality. These findings demonstrate that the compressor design matters in adaptive tokenization.

**Ablation on the adaptive mechanism.** Unifying the router and adaptive compression, we further compare the whole adaptive mechanism of InfoTok to ElasticTok. Specifically, we evaluate both the adaptive mechanism of ElasticTok and InfoTok, to the Cosmos architecture, and a pure ViT architecture. As shown in Table 3 (Right), InfoTok consistently outperforms ElasticTok across both architectures, yielding substantial improvements in both PSNR and FVD. These results confirm the effectiveness and generalization ability of InfoTok.

**Reconstruction under different compression rates.** While we have shown above that an ELBO-based compression rate is theoretically justified and empirically effective, we are curious how the reconstruction of one video changes as we gradually increase/decrease the compression rate. To this end, we visualize in Fig. 3 the reconstruction results with $BPP_{16}$ ranging in $\{0.81, 0.56, 0.31\}$. INFOTOK will reconstruct overall structures of the video when available tokens are limited, such that the MSE loss remains small despite the fine-grained details becoming invisible.

## 5 RELATED WORKS

**Discrete Tokenization** has been extensively explored to facilitate generative AI. Typical VQ-VAE (Van Den Oord et al., 2017) and VQGAN (Esser et al., 2021) extend the Variational Autoencoder (VAE) framework (Kingma et al., 2013) by introducing a learnable codebook to quantize 2D encoded features into discrete tokens, which are then decoded back to reconstruct the original image. To streamline the quantization process, lookup-free tokenization methods such as FSQ (Mentzer et al.,

2023b) and LFQ (Yu et al., 2023c) have been proposed. In terms of representation, TiTok (Yu et al., 2024) employs a Transformer-based architecture to learn 1D tokens. Building upon this, FlowMo (Sargent et al., 2025) and SelfTok (Pan et al., 2025) utilize diffusion decoders conditioned on 1D discrete tokens to mitigate discretization errors further. Along the line of video tokenization, there are several works, including OminiTokenizer (Wang et al., 2024), Open-MAGVIT2 (Luo et al., 2024), and Cosmos (Agarwal et al., 2025). Unlike images, video data is intrinsically more complex and sparse, making existing tokenization approaches inefficient.

**Adaptive Representation** has been studied over the past decade, with early works like Nested Dropout (Rippel et al., 2014) and Matryoshka Representation Learning (Kusupati et al., 2022) introducing methods to learn representations at multiple granularities. Recent approaches have applied similar methodologies to tokenization. CAT (Shen et al., 2025) focuses on continuous tokens, using vision-language models to assess image content complexity heuristically to determine token usage. ALIT (Duggal et al., 2024) recurrently distills 2D tokens into 1D latent tokens, enabling flexible tokenization. One-D-Piece (Miwa et al., 2025) and ElasticTok (Yan et al., 2024) employ random nested dropout strategies for tokens, targeting images and videos, respectively. FlexTok (Bachmann et al., 2025) further incorporates a diffusion decoder to enhance reconstruction quality under low token counts. However, these works focus on images and rely on heuristic methods for adaptive tokenization such as random masking. As such, their objective functions are biased by definition.

## 6 DISCUSSIONS & LIMITATIONS

**Generalizability of INFOTOK.** While this work has concentrated on the video domain, the core principles of INFOTOK are not intrinsically limited to visual sequences. Our framework is founded on information-theoretic compression and utilizes an ELBO-based router to estimate information complexity, a methodology that is broadly applicable to any data modality compatible with a VAE-style reconstruction objective. The fundamental challenge of variable information density exists in many other domains. For instance, audio signals often contain significant temporal redundancy (e.g., periods of silence, sustained background noise), where an adaptive approach could yield substantial compression gains by allocating tokens based on acoustic complexity (Zhang et al., 2023; Liu et al., 2025). Likewise, 3D data, such as point clouds or meshes, often features sparse data and non-uniform detail, with large, simple surfaces requiring less representational capacity than areas of high geometric complexity (Chen et al., 2025; Lu et al., 2025). Therefore, applying INFOTOK to develop adaptive tokenizers for audio, 3D, or other structured data presents a promising direction for future research.

**Limitation of INFOTOK.** While INFOTOK demonstrates significant gains in compression efficiency for video reconstruction, this study has a few limitations. First, our principled ELBO-based router introduces a modest computational overhead by requiring one additional decoder pass to estimate the video's information complexity. Although this is significantly more efficient than the multi-pass search required by prior adaptive methods, further research could explore lighter-weight router mechanisms, such as estimating complexity directly from the encoder's latent representations, to eliminate this extra step entirely. Second, our evaluation is primarily focused on reconstruction fidelity (e.g., PSNR, FVD) as a proxy for representational quality. As noted, we did not extend our experiments to downstream applications such as video generation or action understanding, which was beyond our current scope due to the substantial computational resources required. Investigating how INFOTOK's adaptive token sequences impact the performance and efficiency of large-scale generative or video-understanding models remains a key direction for future work.

## 7 CONCLUSION

This paper introduced InfoTok, a principled adaptive video tokenizer inspired by Shannon's information theory, to address the suboptimality of fixed-rate video compression prevalent in existing methods. InfoTok utilizes an ELBO-based router to determine token lengths dynamically according to video complexity and employs a transformer-based adaptive compressor to manage these variable-length representations efficiently. Empirical results demonstrate InfoTok's significant advantages. We believe that InfoTok offers valuable insights for advancing future research on scalable multimodal models that can both understand and generate long videos.

ETHICS STATEMENT

Video discrete tokenization is pivotal for advancing world models and video generation, offering significant societal benefits by enhancing accessibility and efficiency. However, these advancements also pose ethical challenges, including the potential misuse of generative models to create deceptive content such as deepfakes, which can spread misinformation and infringe on privacy.

REPRODUCIBILITY STATEMENT

We present the full experimental settings in Section 4.1 and provide additional details in Appendix C. To ensure reproducibility, the anonymous code is included in the supplementary material.

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

# Supplementary Material

## CONTENTS

# A    ILLUSTRATION OF ADAPTIVE TOKENIZATION

An illustration of how INFOTOK tokenizes videos based on information complexity (figs. 5 and 6). White tokens are those preserved and black tokens are those masked out by INFOTOK. "Token Usage" specifies how many tokens are preserved in this frame.

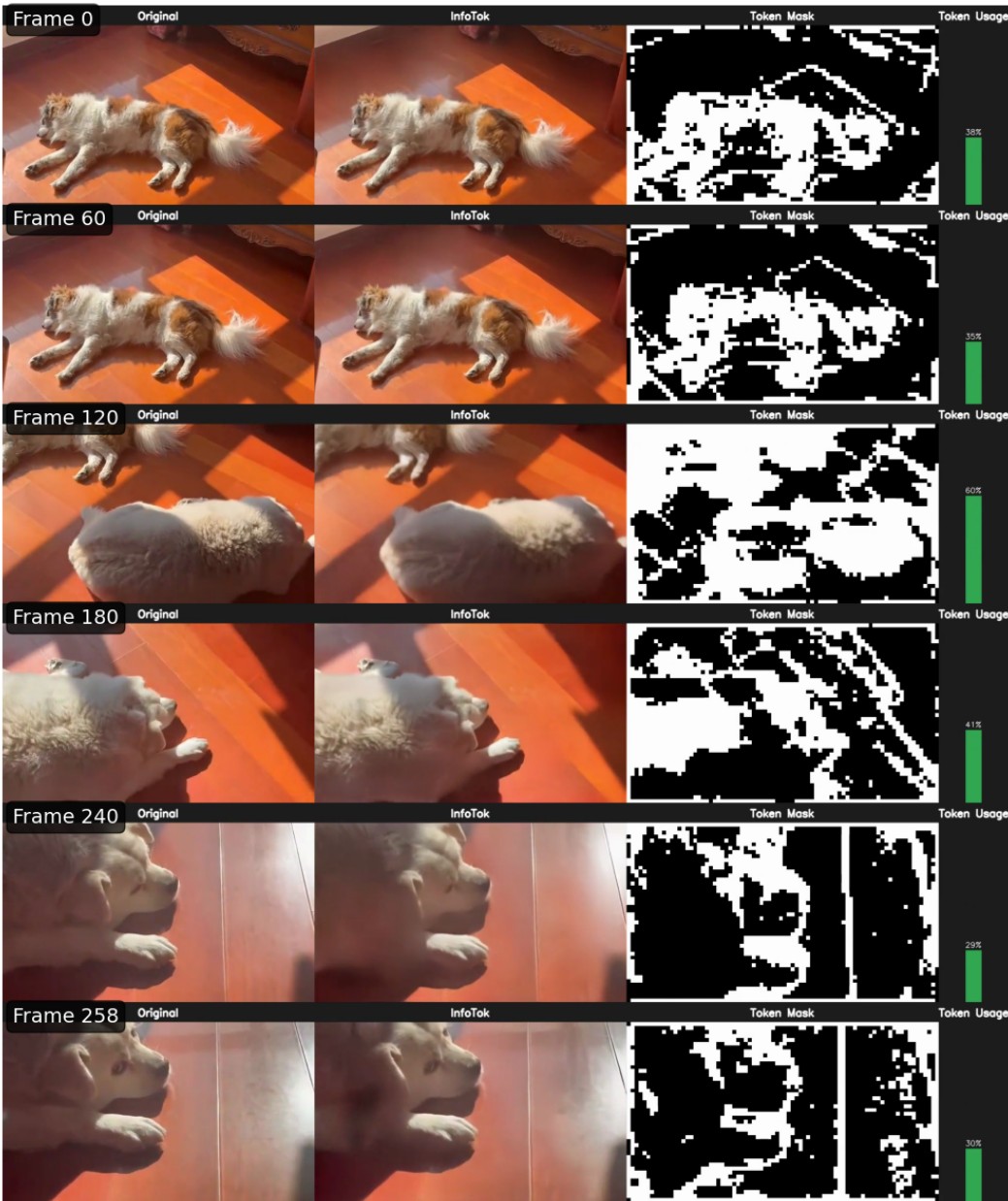

Figure 5: The video starts with a brown dog sleeping on the ground. The video is stable and more than 60% of tokens are masked without harming the reproduction quality. Later on, the camera shifts toward another white dog, during which the token length increases to 60% to encode new information. As the camera stabilizes again and focuses on the face of the white dog, the token length reduces back to 30%, and much of the ground area is masked out (because they can be easily inferred from surrounding areas).

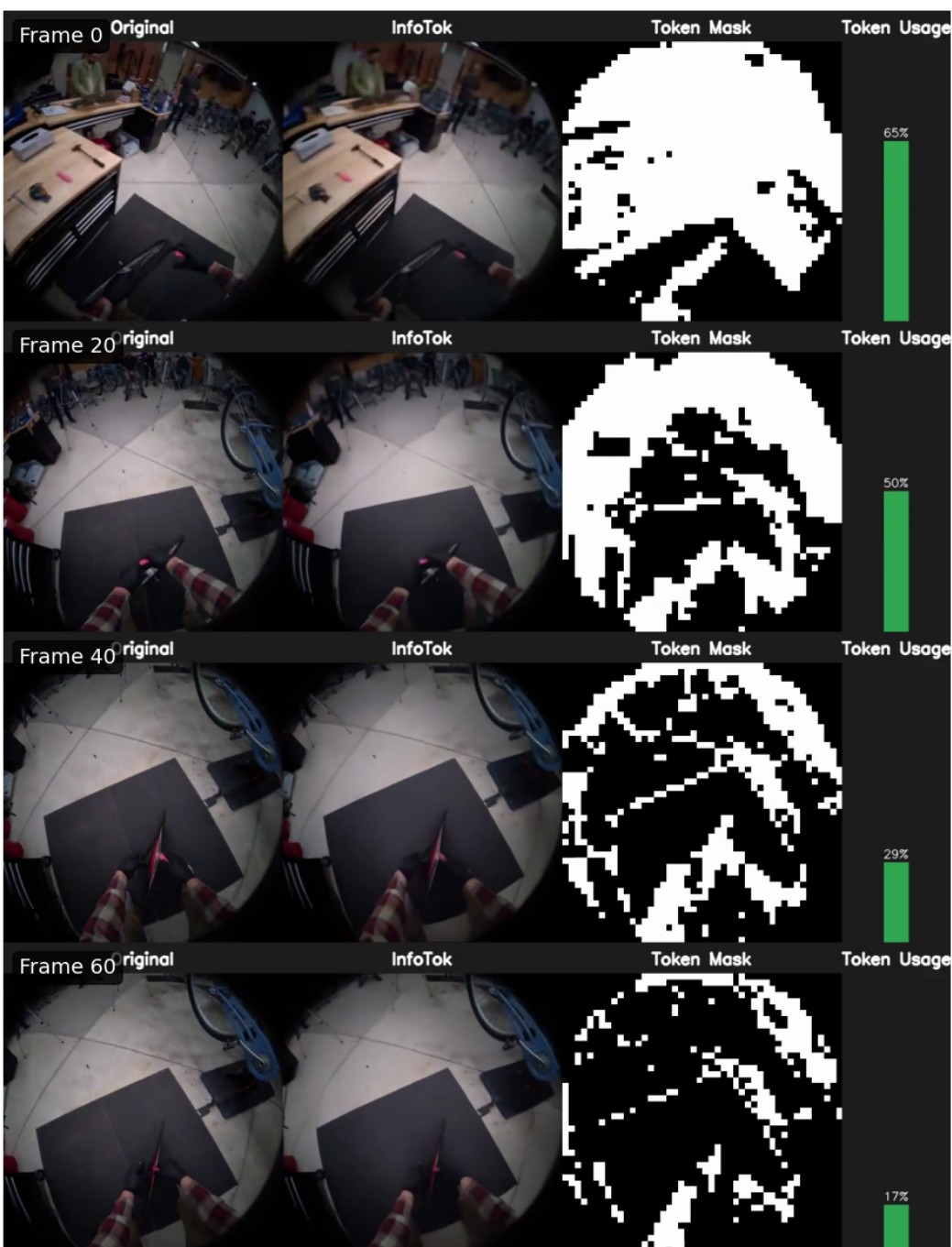

Figure 6: The video begins with a wide shot of a workspace. The camera is highly dynamic as the person moves, requiring a higher token usage of 65% to capture the rapidly changing environment. As the person stabilizes their position and focuses their hands on the task at the center of the frame, the token usage progressively drops to 50% and then 29%. By the end of the sequence, as the background remains static and the movements become more predictable, the token usage further reduces to 17%. Large portions of the peripheral workshop area are masked out, as the model recognizes they contain redundant information that can be inferred from previous frames.

## B  PROOFS

### B.1  PROOF OF THEOREM 2.2

*Proof.* To prove this theorem, we first need to understand the structure of the optimal tokenizer that minimizes eq. (2), as well as the corresponding oracle $r^*$ used during inference.

Given a sampled $\mathbf{x}$ and $N_{\mathbf{x}}$, the loss to be minimized can be represented as

$$\mathbb{E}_{\mathbf{z} \sim q_{\phi,\psi}(\mathbf{z}|\mathbf{x}, N_{\mathbf{x}})}[-\log p_\theta(\mathbf{x}|\mathbf{z})] = \mathbb{E}\Big[-\log p_\theta(\mathbf{x}|\mathcal{Q}(\mathcal{M}_\psi(\mathcal{E}_\phi(\mathbf{x}), N_{\mathbf{x}})))\Big].$$

Due to the compression conducted by $\mathcal{M}_\psi$, multiple videos (that have identical token prefixes) could be mapped to the same sequence, and the decoder is required to minimize the negative log-likelihood for them simultaneously. Therefore, we can rewrite the loss as

$$\mathcal{L} = \mathbb{E}_{\mathbf{z}} \mathbb{E}_{\mathbf{x} \sim p(\mathbf{x}|\mathbf{z})}[-\log p_\theta(\mathbf{x}|\mathbf{z})] \geq \mathbb{E}_{\mathbf{z}} H(P(\mathbf{x}|\mathbf{z})).$$

In other words, given a short token sequence $\mathbf{z}$, the best a decoder can do is to predict $\mathbf{x}$ with probability exactly equaling $p(\mathbf{x}|\mathbf{z})$, in which case the loss is the entropy of the marginal distribution of $\mathbf{x}$ that has token sequence prefixes $\mathbf{z}$, averaged across all possible $\mathbf{z}$ with corresponding probabilities. On the other hand, for an arbitrarily powerful decoder, this loss is realizable. Therefore, below we only need to deal with the converted loss.

Notice that since we assume the compressor follows a generate-then-mask pipeline, the probability that a decoder has input $\mathbf{z}$ is

$$\sum_{\mathbf{x}, |\hat{\mathbf{z}}| = N, \mathbf{z} \subseteq \hat{\mathbf{z}}} p(\mathbf{x}) p(\mathcal{Q}(\mathcal{M}_\psi(\mathcal{E}_\phi(\mathbf{x}), N)) = \hat{\mathbf{z}}) \times \frac{1}{N},$$

where the last $\frac{1}{N}$ comes from the uniform router $r$, and $\mathbf{z} \subseteq \hat{\mathbf{z}}$ means that $\mathbf{z}$ is a prefix of $\hat{\mathbf{z}}$. Notice that since we assume $N$ to be sufficiently large, the probability that two different videos $\mathbf{x}, \mathbf{x}'$ are mapped to an identical $\hat{\mathbf{z}}$ is $0$. Otherwise, we can always find another $|\tilde{\mathbf{z}}| = N$, such that the longest common prefix $\mathbf{z}$ of $\hat{\mathbf{z}}$ and $\tilde{\mathbf{z}}$ is not the prefix of any other video that is not mapped to $\hat{\mathbf{z}}$, i.e. $\nexists \mathbf{x}$ such that, $\mathbf{z} \subseteq \mathcal{Q}(\mathcal{M}_\psi(\mathcal{E}_\phi(\mathbf{x}), N)) \neq \hat{z}$. In this case, we can alter the mapping of $\mathbf{x}'$ to $\tilde{\mathbf{z}}$, and the loss will decrease because the entropy of $P(\mathbf{x}|\mathbf{z})$ for any $\mathbf{z}$ is non-increasing, and for $\hat{\mathbf{z}}$ it strictly decreases.

Therefore, the optimal encoder and compressor will map each $\mathbf{x}$ to a different $\hat{\mathbf{z}}$ with length $N$, and mask $\hat{\mathbf{z}}$ according to the router's sample $N_{\mathbf{x}}$. We denote the mapping from $\mathbf{x}$ to $\hat{\mathbf{z}}$ as $M(\mathbf{x}) = \mathcal{Q}(\mathcal{M}_\psi(\mathcal{E}_\phi(\mathbf{x}), N))$. Given a masked token sequence $\mathbf{z}$ with length $N_{\mathbf{x}}$ sampled by the router, the decoder will predict $\mathbf{x}$ with probability proportional to $\mathbb{I}[\mathbf{z} \subseteq M(\mathbf{x})]p(\mathbf{x})$.

We can therefore model the entire tokenizer as a $C$-fold tree with depth $N$, where each leaf corresponds to a video $\mathbf{x}$ with probability $p(\mathbf{x})$ or corresponds to nothing. For each video $\mathbf{x}$, there is one and only one path from the root node to it, representing the (uncompressed) token sequence of the video. The depths of leaf nodes are video lengths, which are defined to be the minimal token lengths such that no two videos are mapped to the same node. The entropy of all leaf nodes is $0$, and the entropy of an intermediate node (corresponding to a compressed token sequence $\mathbf{z}$) is the entropy of the marginal distribution over all videos with prefixes $\mathbf{z}$. The loss can be presented as

$$\mathcal{L} = \sum_{\mathbf{z}} \sum_{\mathbf{z} \subseteq M(\mathbf{x})} \frac{p(\mathbf{x})}{N} \cdot H(P(\cdot|\mathbf{z}))$$

$$= \sum_{\mathbf{z}} \sum_{\mathbf{z} \subseteq M(\mathbf{x})} \frac{p(\mathbf{x})}{N} \cdot \Big( \sum_{\mathbf{z} \subseteq M(\mathbf{x})} -\log \frac{p(\mathbf{x})}{\sum_{\mathbf{z} \subseteq M(\tilde{\mathbf{x}})} p(\tilde{\mathbf{x}})} \cdot \frac{p(\mathbf{x})}{\sum_{\mathbf{z} \subseteq M(\tilde{\mathbf{x}})} p(\tilde{\mathbf{x}})} \Big).$$

In the proof below, we can simply assume that $C = 2$, but the proof can be trivially extended to any general $C$.

First, we rewrite the loss $\mathcal{L}$ to get an equivalent loss $L(T) = N\mathcal{L}$ for a tree $T$. For each leaf node $i$ (corresponding to video $\mathbf{x}$), define $N(i)$ as the set of ancestors of $i$ (corresponding to $M(\mathbf{x})$), and define $N_i$ as the set of leaf nodes that have node $i$ as an ancestor. Consequently, define

$$H(N_j) = \sum_{i: j \in N(i)} -\frac{p(i)}{p(j)} \log \frac{p(i)}{p(j)}.$$

Here, $p(j)$ is the sum of probability for an intermediate node $j$, which equals to $\sum_{\mathbf{z} \subseteq M(\tilde{\mathbf{x}})} p(\tilde{\mathbf{x}})$ in the previous definition. Therefore, we have

$$
\begin{aligned}
L(T) =& N \sum_i p(i) \sum_{j \in N(i)} \frac{1}{N} \sum_{i:j \in N(i)} -\frac{p(i)}{p(j)} \log \frac{p(i)}{p(j)} \\
=& N \sum_i p(i) \sum_{j \in N(i)} \frac{1}{N} H(N_j) \\
=& \sum_j \sum_{i:j \in N(i)} p(i) H(N_j) \\
=& \sum_j p(j) H(N_j) \\
=& \sum_j p(j) \sum_{i:j \in N(i)} -\frac{p(i)}{p(j)} \log \frac{p(i)}{p(j)} \\
=& \sum_j \sum_{i:j \in N(i)} -p(i) \log \frac{p(i)}{p(j)} \\
=& \sum_j \left[ p(j) \log p(j) - \sum_{i:j \in N(i)} p(i) \log p(i) \right] \\
=& \sum_j p(j) \log p(j) - \sum_i l(i) p(i) \log p(i) \\
=& \sum_j l(j) \left[ p(j) \log p(j) - p(2j) \log p(2j) - p(2j+1) \log p(2j+1) \right] \\
=& \sum_j p(j) l(j) \left( -\frac{p(2j)}{p(j)} \log \frac{p(2j)}{p(j)} - \frac{p(2j+1)}{p(j)} \log \frac{p(2j+1)}{p(j)} \right) \\
=& \sum_j p(j) l(j) H(j),
\end{aligned}
$$

where $j$ is all intermediate nodes, $p(j)$ is the sum of probabilities of its children, $l(j)$ is the depth of the node (the root node has depth 0), and $H(j)$ is the entropy of the node, i.e.

$$
H(j) = \text{Entropy}\left( \frac{p(2j)}{p(2j) + p(2j+1)}, \frac{p(2j+1)}{p(2j) + p(2j+1)} \right).
$$

Here $2j$ and $2j+1$ are the left and right child of node $j$,

$$
\text{Entropy}(a, b) = -(a \log_2 a + b \log_2 b).
$$

To better understand the loss, we further separate it in terms of depth:

$$
L(T) = \sum_{d \geq 1} df_d, \tag{5}
$$

where $d$ represents the depth, and

$$
f_d = \sum_{j:l(j)=d} p(j) H(j).
$$

Since $H(j) \in [0, 1]$, we have $0 \leq f_d \leq \sum_{j:l(j)=d} p(j) \leq 1$.

Further, an important property here is that

$$
\begin{aligned}
\sum_d f_d &= \sum_d \sum_{j:l(j)=d} p(j)H(j) \\
&= \sum_j p(j)H(j) \\
&= \sum_j -p(j)\left(\frac{p(2j)}{p(2j)+p(2j+1)}\log\frac{p(2j)}{p(2j)+p(2j+1)} + \frac{p(2j+1)}{p(2j)+p(2j+1)}\log\frac{p(2j+1)}{p(2j)+p(2j+1)}\right) \\
&= \sum_j \left(p(2j)+p(2j+1)\right)\log\left(p(2j)+p(2j+1)\right) - p(2j)\log p(2j) - p(2j+1)\log p(2j+1) \\
&= \sum_{i:i \text{ is a leaf node.}} -p(i)\log p(i) \\
&= H_2(\mathbb{D}).
\end{aligned}
$$

Now we start to prove the main theorem. We let the data distribution $\mathbb{D}$ be as the distribution for $2^M$ videos, with $p(j) = 2^{-j}$ for $j = 1, \ldots, 2^M - 1$, and $p(2^M) = p(2^M - 1)$. We aim to prove that the expected depth of the tree $T_M$ minimizing the loss 5 under $\mathbb{D}$ goes to infinity as $M$ goes to infinity.

We prove by contradiction. If the expected depth doesn't go to infinity as $M \to \infty$, then there exists $D > 0$ such that $l(1) \leq D$ for any sufficiently large $M$. In this case, we claim that we can find a tree that yields a smaller loss, by "lifting" a node $j$ with sufficiently small probability and sufficiently deep depth up to be a sibling of node 1. Specifically, suppose node 1 was at depth $d_1$, it will now be moved one level down to depth $d_1 + 1$, with node $j$ as its sibling. The previous sibling of node $j$ will be moved one level up. This $j$ always exists since $m \to \infty$, and the maximum depth of the tree will increase to infinity simultaneously. Specifically, we choose node $j$ such that it has depth larger than $2D$, and $p_j$ is sufficiently small compared with all $p_i$ for $i$ with depth less than or equal to $2D$.

Now, let's look at the change to $f_d$ after the lifting. Without loss of generality, we assume the root is the lowest common ancestor of node 1 and node $j$. If not, we can find their lowest common ancestor and do the same analysis on the sub-tree with that ancestor as the root. We assume node 1 is in the left sub-tree, and node $j$ was originally in the right sub-tree. We decompose the change to $f_d$ into four different parts.

**Part one. $\Delta_1$: change to $f_{d_1}$ in the left sub-tree.**

The branching of node 1 will add an extra term $(p(1) + \Delta_p)\,\mathrm{Entropy}(\frac{p(1)}{p(1)+\Delta_p}, \frac{\Delta_p}{p(1)+\Delta_p})$ to $f_{d_1}$, formalized below:

$$
\begin{aligned}
\Delta_1 &= (p(1)+\Delta_p)\,\mathrm{Entropy}\left(\frac{p(1)}{p(1)+\Delta_p}, \frac{\Delta_p}{p(1)+\Delta_p}\right) \\
&= -p(1)\log\frac{p(1)}{p(1)+\Delta_p} - \Delta_p\log\frac{\Delta_p}{p(1)+\Delta_p} \\
&= -p(1)\log p(1) + p(1)\log(p(1)+\Delta_p) - \Delta_p\log\Delta_p + \Delta_p\log(p(1)+\Delta_p) \\
&= -p(1)\log p(1) + p(1)\left(\log p(1) + \frac{\Delta_p}{p(1)}\right) - \Delta_p\log\Delta_p + \Delta_p\left(\log p(1) + \frac{\Delta_p}{p(1)}\right) + o(\Delta_p) \\
&= -\Delta_p\log\Delta_p + o(\Delta_p)
\end{aligned}
$$

**Part two. $\Delta_2$: change to $f_d$ in the left sub-tree with $0 < d < d_1$.**

For each $d < d_1$, suppose the ancestor of node 1 at level $d$ originally has two sub-trees with probabilities $p_1$ and $p_2$. After the operation, these two probabilities become $p_1 + \Delta_p$ and $p_2$ respectively. Hence, the change at level $d$ is:

$$\Delta_3 = (p_1 + p_2 + \Delta_p)\text{Entropy}\left(\frac{p_1 + \Delta_p}{p_1 + p_2 + \Delta_p}, \frac{p_2}{p_1 + p_2 + \Delta_p}\right) - (p_1 + p_2)\text{Entropy}\left(\frac{p_1}{p_1 + p_2}, \frac{p_2}{p_1 + p_2}\right)$$

$$= (p_1 + p_2 + \Delta_p)\log(p_1 + p_2 + \Delta_p) - (p_1 + \Delta_p)\log(p_1 + \Delta_p)$$
$$\quad - p_2 \log p_2 - (p_1 + p_2)\log(p_1 + p_2) + p_1 \log p_1 + p_2 \log p_2$$

$$= (p_1 + p_2 + \Delta_p)\left(\log(p_1 + p_2) + \frac{\Delta_p}{p_1 + p_2}\right) - (p_1 + \Delta_p)\left(\log p_1 + \frac{\Delta_p}{p_1}\right)$$
$$\quad - p_2 \log p_2 - (p_1 + p_2)\log(p_1 + p_2) + p_1 \log p_1 + p_2 \log p_2 + o(\Delta_p)$$

$$= \Delta_p \log \frac{p_1 + p_2}{p_1} + o(\Delta_p)$$

$$= O(\Delta_p)$$

Since $d < d_1$ is upper bounded by $D$, by summing up change at all level $d$, we conclude that $\Delta_2 = O(\Delta_p)$

**Part three.** $\Delta_3$: **change to** $f_d$ **at level 0.** Suppose the root originally has two sub-trees with probabilities $p_1$ and $p_2$. After the operation, these two probabilities become $p_1 + \Delta_p$ and $p_2 - \Delta_p$ respectively. Hence, the change at level 0 is

$$(p_1 + p_2)\text{Entropy}\left(\frac{p_1 + \Delta_p}{p_1 + p_2}, \frac{p_2 - \Delta_p}{p_1 + p_2}\right) - (p_1 + p_2)\text{Entropy}\left(\frac{p_1}{p_1 + p_2}, \frac{p_2}{p_1 + p_2}\right)$$

$$= (p_1 + p_2)\log(p_1 + p_2) - (p_1 + \Delta_p)\log(p_1 + \Delta_p)$$
$$\quad - (p_2 - \Delta_p)\log(p_2 - \Delta_p) - (p_1 + p_2)\log(p_1 + p_2) + p_1 \log p_1 + p_2 \log p_2$$

$$= (p_1 + p_2)\log(p_1 + p_2) - (p_1 + \Delta_p)\left(\log p_1 + \frac{\Delta_p}{p_1}\right)$$

$$\quad - (p_2 - \Delta_p)\left(\log p_2 - \frac{\Delta_p}{p_2}\right) - (p_1 + p_2)\log(p_1 + p_2) + p_1 \log p_1 + p_2 \log p_2 + o(\Delta_p)$$

$$= \Delta_p \log \frac{p_2}{p_1} + o(\Delta_p)$$

$$= O(\Delta_p)$$

**Part four.** $\Delta_4$: **change to** $f_d$ **in the right sub-tree with** $0 < d < 2D$.

Similar to Part two, $\Delta_4 = O(\Delta_p)$.

**Part five.** $\Delta_5$: **change to** $f_d$ **in the right sub-tree with** $d \geq 2D$.

Suppose node $j'$ is node $j$'s ancestor in the right sub-tree at level $2D$. Suppose the node $j'$ originally has probability $p_0$. After the operation, it has probability $p_0 - \Delta_p$. Similar to the derivation of $\sum_d f_d$, we can calculate that

$$\Delta_5 = (p_0 - \Delta_p)\log(p_0 - \Delta_p) - (p_0 \log p_0 - \Delta_p \log \Delta_p)$$

$$= (p_0 - \Delta_p)\left(\log p_0 - \frac{\Delta_p}{p_0}\right) - p_0 \log p_0 + \Delta_p \log \Delta_p + o(\Delta_p)$$

$$= \Delta_p \log \Delta_p + O(\Delta_p)$$

By all above, since $\Delta_p$ is sufficiently small, the only dominant terms among the $\Delta$'s are the increase of $-\Delta_p \log \Delta_p$ at level $d_1$, and decrease of $\Delta_p \log \Delta_p$ at level $2D$ and below.

Recall that $L(T) = \sum_{d \geq 1} d f_d$, and $2D > D > d_1$, we'll have loss decrease by at least $-(2D - d_1)\Delta_p \log \Delta_p$ after the operation. This contradicts our assumption that $T_M$ is the minimizer of the loss. Therefore, we have the expected depth of the $T_M$ minimizing loss 5 under $\mathbb{D}$ goes to infinity as $M$ goes to infinity.

On the other hand, we can construct Hoffman tree with node $j$ on level $j$. In this case,

$$\lim_{M \to \infty} \sum_{i=1}^{2^M - 1} i 2^{-i} + 2^M 2^{-(2^M - 1)} = 2.$$

Hence, $H_2(\mathbb{D}) = O(1)$. This means that for any $\kappa > 0$, there exists $\mathbb{D}$ and sufficiently large $N$ such that $\mathbb{E}_{\mathbf{x} \sim p(\mathbf{x}), N_{\mathbf{x}} \sim r^*(N_{\mathbf{x}}|\mathbf{x})}[N_{\mathbf{x}}] \geq \kappa H_C(\mathbb{D})$. This completes the proof.

$\square$

### B.2 Proof of Theorem 3.1

*Proof.* Suppose the ELBO for $\mathbf{x}$ is $\mathrm{ELBO}(\mathbf{x}) = l_{\mathbf{x}} \leq \log p(\mathbf{x})$. Since $-l_{\mathbf{x}} \geq -\log p(\mathbf{x})$, and Hoffman tree can encode $\mathbf{x}$ with $N_{\mathbf{x}} = -\log p(\mathbf{x})$, the minimizer of the adaptive loss has $N_{\mathbf{x}} \leq -l_{\mathbf{x}}$ with $N \geq -l_{\mathbf{x}}$.

Therefore, we have

$$
\begin{aligned}
\mathbb{E}_{\mathbf{x} \sim p(\mathbf{x}), N_{\mathbf{x}} \sim r(N_{\mathbf{x}}|\mathbf{x})}[N_{\mathbf{x}}] \leq & \mathbb{E}_{\mathbf{x} \sim p(\mathbf{x}), N_{\mathbf{x}} \sim r(N_{\mathbf{x}}|\mathbf{x})} - l_{\mathbf{x}} \\
= & H_C(\mathbb{D}) + \mathbb{E}_{\mathbf{x} \sim p(\mathbf{x}), N_{\mathbf{x}} \sim r(N_{\mathbf{x}}|\mathbf{x})} \left( -l_{\mathbf{x}} + \log p(\mathbf{x}) \right) \\
= & H_C(\mathbb{D}) - \mathbb{E}[\mathrm{ELBO}(\mathbf{x})] - \mathbb{E}_{\mathbf{x} \sim p(\mathbf{x})}[-\log p(\mathbf{x})] \\
\leq & H_C(\mathbb{D}) + \beta - \mathbb{E}_{\mathbf{x} \sim p(\mathbf{x})}[-\log p(\mathbf{x})].
\end{aligned}
$$

$\square$

## C Experimental Details

### C.1 Training Details.

**Architecture Configuration.** The complete hyperparameter settings of the INFOTOK model architecture are provided in Table 4. The adaptive compressor and de-compressor are implemented using a standard Vision Transformer (ViT) equipped with 2D Rotary Position Embeddings (RoPE)Heo et al. (2024) to handle videos with different resolutions. Features obtained from the Cosmos encoder are processed by an eight-layer ViT encoder and subsequently compressed using ELBO-guided length selection and likelihood-based token selection, as described in Sections 2 and 3.2. The compressed discrete tokens are decoded back into continuous features via an eight-layer ViT decoder and passed through the Cosmos Tokenizer's decoder to reconstruct the original video. Despite its effectiveness, the adaptive compressor/de-compressor module constitutes only 14.6% of the total parameters of INFOTOK, with the base tokenizer accounting for the remaining 85.4%. This highlights the efficiency and potential of our adaptive tokenization framework.

**Training Datasets.** For training, we use the datasets in Agarwal et al. (2025), in coverage of different visual objects and multiple resolution. During training, to compare fairly with ElasticTok Yan et al. (2024), we train our models on resized videos with 256px (e.g., 256×256, 256×456 and so on), although our method works naturally on different resolutions.

**Training Configuration and Computation Resource.** The training augmentation follows Cosmos Tokenizer Agarwal et al. (2025). We employ the AdamW Loshchilov & Hutter (2017) optimizer with an initial learning rate $1e-4$ with cosine decay to $1e-5$ for $1e5$ steps. The training regime features a batch size of 1 with video clips of 33 frames over $2e5$ steps. All models presented in this paper are trained with 32 H100 GPUs (4 nodes), and our setting will take 4 days to train a model. Notice that each GPU can consume data with different resolution simultaneously.

### C.2 Implementation Details

**Token length selection.** To implement ELBO-based token length selection, we address the challenge of computing the expectation of the Evidence Lower Bound (ELBO) in the context of streaming large-scale video datasets, where precomputing ELBO values for all data is impractical. To overcome this, we maintain an exponentially weighted moving average (EMA) of the ELBO over incoming training data, providing an efficient approximation of its expectation. This approach allows for real-time adaptation to the data distribution without the need for full dataset traversal. After that, as shown in Table 4, for the INFOTOK model, we rescale the ratio $\frac{\mathrm{ELBO}(\mathbf{x})}{\mathbb{E}[\mathrm{ELBO}(\mathbf{x})]}$ by a fixed compression factor $\beta$ (e.g., 0.5) to determine the final token length. Conversely, in the INFOTOK-Flex variant,

Table 4: Architecture Configuration.

| Hyperparameter | InfoTok-$\beta$ | InfoTok-Flex |
|---|---|---|
| **Adaptive Compressor/De-compressor** | | |
| Patch size | 1 | - |
| Encoder depth | 8 | - |
| Decoder depth | 8 | - |
| Attention head | 32 | - |
| Token hidden size | 256 | - |
| MLP hidden size | 512 | - |
| Positional embedding | 2D RoPE Heo et al. (2024) | - |
| Compression Factors $\mathcal{B}$ | $[\beta]$ | $[0.25, 0.5, 0.75, 1]$ |
| Parameter size | 18M (14.6%) | - |
| **Quantizer** | | |
| Model type | FSQ Mentzer et al. (2023b) | - |
| Embedding size | 6 | - |
| Feature level | [8, 8, 8, 5, 5, 5] | - |
| Codebook size | 64000 ($2^{16}$) | - |
| **Based Tokenizer: Cosmos Agarwal et al. (2025)** | | |
| Model type | 3D Causal CNN | - |
| Temporal-spatial compression | (4, 8, 8) | - |
| Parameter size | 105M (85.4%) | - |
| Total parameter size | 123M | - |

we introduce additional flexibility by randomly sampling $\beta$ from the set $[0.25, 0.5, 0.75, 1]$ for each incoming data instance. INFOTOK-Flex enables the model to ensemble different compression levels, enhancing its adaptability to varying content complexities. We clip the minimal token length of each video to $1/16$ of maximal token length to avoid over-compression and instable training.

**Adaptive Compressor.** As detailed in Section 3.2, we implement ELBO-guided token pruning by removing tokens with the highest log-likelihoods, corresponding to the lowest information content. This process compresses the discrete token sequence to the length specified by the ELBO-based token length selection. To achieve this efficiently, we reuse the pixel-wise ELBO values computed during the length selection phase. These values are aggregated at the patch indicated by compression level (e.g., (4, 8, 8) for Cosmos Base Tokenizer) and mapped back to their corresponding discrete tokens which are temporal and spatial aligned. Notably, this approach introduces negligible computation overhead and facilitates efficient token compression.

## C.3 EVALUATION DETAILS

**Inference Pipeline.** For a long video, we separate it as several clips of 33 frames, and reconstruct them one-by-one. During inference, given the average compression rate (measure by $\text{BPP}_{16}$), we firstly determine the average token length used for a video clip of 33 frames. For instance, given a clip in shape of $(33, 256, 256)$ and targeting $\text{BPP}_{16}$ of 0.5625, the maximum token length is $9216 = 9 \times 32 \times 32$, the average token length is $4608 = 9216 \times (0.5625 - 0.0625)$, where 0.0625 is additional bits carrying the token location information. Then the desired token length of a certain video clip $x$ is $4608 \times \frac{\text{ELBO}(\mathbf{x})}{\mathbb{E}[\text{ELBO}(\mathbf{x})]}$, where $\mathbb{E}[\text{ELBO}(\mathbf{x})]$ is approximated by computing the average ELBO of the provided evaluation dataset during inference. Additionally, if there is only limited data provided, it can be directly approximated by $\text{ELBO}(\mathbf{x})_{\text{ema}}$, which is the exponentially weighted moving average of ELBO obtained during training with smoothing factor as 0.99, as mentioned in Section C.2.

**Dataset Details.** TokenBench Agarwal et al. (2025) is a curated benchmark designed to standardize the evaluation of video tokenizers. It comprises 500 high-resolution, long-duration videos sampled from diverse datasets, including BDD100K Yu et al. (2020), EgoExo-4D Grauman et al. (2024), BridgeData V2 Walke et al. (2023), and Panda-70M Chen et al. (2024). These videos encompass a

Table 5: Comparison of Cosmos Arch with and without InfoTok across resolutions on TokenBench.

| Model | Resolution | $BPP_{16}$ | PSNR | FVD |
|---|---|---|---|---|
| Cosmos Arch | $256 \times 256$ | 1.00 | 30.01 | 49 |
| Cosmos Arch + InfoTok | $256 \times 256$ | 0.56 | 29.27 | 70 |
| Cosmos Arch | 360p (1:1, 4:3, 16:9) | 1.00 | 31.13 | 27 |
| Cosmos Arch + InfoTok | 360p (1:1, 4:3, 16:9) | 0.56 | 30.55 | 56 |

Table 6: Inference latency comparison across different methods.

| Method | NFEs | Inference Latency per video |
|---|---|---|
| Cosmos | 1 | 0.61 s |
| Cosmos + InfoTok mechanism | 2 | 1.23 s |
| Cosmos + ElasticTok mechanism | 12 | 13.45 s |
| ElasticTok | 12 | 42.75 s |

wide range of complex scenarios such as autonomous driving, egocentric activities, robotic manipulation, and web videos, providing a comprehensive evaluation suite for video tokenization methods. For the DAVIS dataset Caelles et al. (2019), we utilize the Test-Dev 2019 split, which consists of 30 video sequences featuring various human activities. As mentioned in Section 4.1, all videos are resized and cropped into 256px to ensure fair comparison with baselines Yan et al. (2024) only supporting square video data tokenization.

## D ADDITIONAL EXPERIMENT RESULTS

**Ablation on different resolutions.** We further evaluate InfoTok across multiple resolutions to assess its generalization ability, as shown in Table 5. The results show a consistent trend across diverse aspect ratios at higher resolutions, indicating that InfoTok generalizes well and adaptively tokenizes videos under varying resolution settings.

**Inference latency comparison.** We also compare the inference latency of InfoTok and ElasticTok at the same compression rate ($BPP_{16} = 0.56$) on a single NVIDIA RTX A5000 GPU, using videos of 33 frames at a resolution of $256 \times 256$. As shown in Table 6, InfoTok achieves significantly lower inference latency compared to ElasticTok.

## E USE OF LARGE LANGUAGE MODELS

We use large language models to polish our writing, e.g., to check grammar and improve paper fluency.

