# OpenReview forum: "InfoTok: Adaptive Discrete Video Tokenizer via Information-Theoretic Compression"
_ICLR.cc/2026/Conference — ICLR 2026 Oral_

### Official Review · Reviewer_66nG · 2025-10-31

**Soundness:** 3
**Presentation:** 3
**Contribution:** 3
**Rating:** 8
**Confidence:** 4

**Summary:**

This paper proposes an adaptive visual tokenizer that allocates a more optimal compressed sequence length than existing methods based on information theoretic principles. This paper demonstrates that prior works’ training methods are often suboptimal, imputing bias that makes the adaptive compressions lower-quality despite their adaptability. It further claims that the optimal design for the compressed token length should be proportional to the negative log likelihood of the input, and thus uses the learned ELBO to modulate the input compression factor. The proposed router uses this principle to more optimally allocate tokens for compression, and the top K tokens are selected based on their log likelihood. The results demonstrate that InfoTok achieves better performance at the same compression rates as comparable methods.

**Strengths:**

1) This paper provides some much-needed theoretical grounding to the adaptive tokenizer space, and attempts to pin down what `optimal’ adaptive compression should look like.

2) Experiments are very comprehensive and clearly demonstrate that InfoTok achieves better reconstruction at the same compression rate compared to other adaptive tokenizers.

3) While the results are incremental on their own, the entire paper is well structured and motivated, and properly places all the contributions in context, making the result valuable for the community.

4) While the claims in the theorems are initially hard to parse, they are empirically supported by experiments that make the intuition clearer.

**Weaknesses:**

1) Key definitions are not well highlighted in the text, and the mathematical sections are poorly written and hard to follow. Since definitions are missing from the theorems, the explanation is rather hard to follow. For example, the entropy H is not clearly defined, \mathbb{D} not defined. Similarly r(N | x) not defined in Alg 1 input. While I may have missed these definitions, I would suggest not burying them in the text and making key components clear and easier for readers to refer to.

2) It would be helpful if there were some visual results demonstrating the differences in InfoTok's reconstruction at different token lengths for the same image, so we could see what inputs are determined to need lower ones and which are determined to need higher ones. The reconstructions shown in Figure 2 are not particularly compelling - the compression results are considerably worse than Cosmos-DV.

3) There are no video samples provided in the supplementary material or project page which makes qualitative analysis of the results challenging.

**Questions:**

I don't have any strong suggestions, but would appreciate responses to my above critiques and would like to see them addressed in the next version of the paper.

---

> ### Author Response · Authors · 2025-11-16
>
> We sincerely thank the reviewer for the recognition of our contributions, and the careful reviews that can help us further improve our work. Below we address your concerns separately.
> - [Highlight of the notation] We thank the reviewer for pointing out the issue that some definitions are cannot be easily found. Specifically, the entropy $H$ is defined in Theorem 2.1 (Line 148); $\mathbb D$ is defined in Line 128 as the video distribution, i.e. the distribution that video $\mathbf x$ belongs to; and $r(N|x)$ is explicitly defined in the input of Alg. 1 as "the router $r(\cdot)$". In the modified version, we have make all these definitions more clear and we mark our changes red.
> - [InfoTok's reconstruction at different token lengths.] We thank the reviewer for the visualization suggestion. We have added a new figure showing the reconstruction results of the same video under different rate in the paper (Figure 3), as well as a paragraph analyzing this (we copy it below for your reference). Notice that our original comparison with Cosmos-DV (in Figure 2) is under different compression rates: we achieve similar result with only **a half** of tokens. With slighly more tokens, our PSNR quickly becomes higher than Cosmos-DV. Please let us know if there is any additional visualization you would like to see.
> > While we have shown above that an ELBO-based compression rate is theoretically justified and empirically effective, we are curious how the reconstruction of one video changes as we gradually increase/decrease the compression rate. To this end, we visualize in Fig.~\ref{fig:bpp_vis} the reconstruction results with BPP$_{16}$ ranging in $\{0.81, 0.56,0.31\}$. \alg will reconstruct overall structures of the video when available tokens are limited, such that the MSE loss remains small despite the fine-grained details becoming invisible.
> - [Visualization of videos] Thanks for your suggestion. We have uploaded the corresponding videos (comparison of different methods and different rates) in https://gofile.io/d/TIjsoC. We hope that help address your concern.
>
> Again, we thank the reviewer for the recognition of our work, and please let us know if there is any additional question we can help address.

---

> > ### Comment · Reviewer_66nG · 2025-11-24
> > **Response to authors.**
> >
> > Thanks, this addressed all my concerns. I will keep my rating at 8 for now, but may increase it pending resolution of other author's concerns.

---

### Official Review · Reviewer_t2hE · 2025-11-01

**Soundness:** 3
**Presentation:** 3
**Contribution:** 3
**Rating:** 6
**Confidence:** 3

**Summary:**

This paper proposes InfoTok, an adaptive discrete tokenizer that leverages an information-theoretic formulation to allocate token budgets based on the data's compressibility. The method introduces a principled way to determine token lengths using a normalized ELBO-based router and achieves strong empirical performance with a conceptually simple approach.

**Strengths:**

* The paper is well structured, and the mathematical formulation is precise.
* It achieves competitive or superior performance using a lightweight and interpretable mechanism.
* The authors provide rigorous justifications connecting ELBO with optimal token length, and the derivation is insightful.
* The ELBO-based routing and token selection reuse existing encoder-decoder structures and introduce minimal inference cost, which is appealing in practical deployments.

**Weaknesses:**

* Is the per-token ELBO computed purely based on the encoder-decoder’s end-to-end reconstruction path? Must this be explicitly introduced during training, or can the method be plugged into any VAE-style model without changes? Specifically, can non-VAE tokenizers be adapted to this framework?
* If N_max is small or the compression ratio is extremely low (e.g., β< 0.1), what are the observed effects on stability and convergence? Would the KL term dominate or explode in such settings, and does it impact model converging or generalization?
* The method is designed for videos. Would it generalize to other structured data types such as audio and 3D point clouds? Do different modalities affect the ELBO-based router's assumptions or effectiveness?

**Questions:**

Please refer to the weaknesses.

---

> ### Author Response · Authors · 2025-11-16
>
> We sincerely thank the reviewer for the careful and constructive review. We are happy that the reviewer thinks our paper is "well structure, precise, insightful, practical, and competitive". Below we address your concerns separetely.
> - [ELBO computation] We thank the reviewer for these clarifying questions. InfoTok consists of two key components: the router and the adaptive compressor. The ELBO estimate used by the router is indeed computed using the VAE tokenizer. While we use the base VAE's structure to naturally compute this ELBO-based length, the router itself could theoretically be a separate, off-the-shelf model that estimates information complexity without training. However, the adaptive compressor ($M_{\psi}$) and de-compressor ($M_{\psi}^{-1}$) are new modules that must be explicitly trained, as detailed in Algorithm 1. This is because a standard tokenizer's decoder is not designed to reconstruct from the variable-length, compressed representations our method produces.
> - [Non-VAE tokenizers] The adaptive compressor component itself is general and not tied to a specific architecture. If the reviewer is referring to standard Autoencoders (AEs) as non-VAE tokenizers, then yes, it is possible to adapt them into an adaptive counterpart. In such a scenario, one would need an additional router to determine the token lengths ($N_x$). We chose a VAE-style tokenizer (Cosmos 5) in our work because it conveniently and naturally serves as both the base architecture and the router (via its ELBO estimate 6), simplifying the overall design. Please do not hesitate to let us know if we misunderstand your question and would like further explnanation.
> - [Infotok under extreme case] In practical implementation, we preset a lower bound of the number of tokens ($\text{BPP}_{16} \geq 1/16$) to ensure that InfoTok can provide reasonable results. If we use a much smaller $\beta$, the reconstructed videos will contain mostly the global features and miss the details. In addition to your question, we attached the reconstruction result of the same video under different rate in the paper (new Figure 3 and a paragraph of description) for you reference. As you can see, the high-to-low frequency information appears gradually.
> - [Generalizability of InfoTok] We thank the reviewer for pointing out this. Generalizing Infotok to other domains such as images, audio, and 3D point clouds is definitely possible, since the mathematical structure and assumptions of InfoTok is not specifically tie to videos. We have included an additional paragraph in the revised paper about generalizing InfoTok to different domains, and we attach it here for your reference.
> > While this work has concentrated on the video domain, the core principles of InfoTok are not intrinsically limited to visual sequences. Our framework is founded on information-theoretic compression and utilizes an ELBO-based router to estimate information complexity, a methodology that is broadly applicable to any data modality compatible with a VAE-style reconstruction objective. The fundamental challenge of variable information density exists in many other domains. For instance, audio signals often contain significant temporal redundancy (e.g., periods of silence, sustained background noise), where an adaptive approach could yield substantial compression gains by allocating tokens based on acoustic complexity. Likewise, 3D data, such as point clouds or meshes, often features sparse data and non-uniform detail, with large, simple surfaces requiring less representational capacity than areas of high geometric complexity. Therefore, applying InfoTok to develop adaptive tokenizers for audio, 3D, or other structured data presents a promising direction for future research.
>
> Again, we thank the reviewer for the recognition of our work, and please let us know if there are any additional questions we can help address.

---

### Official Review · Reviewer_5w2J · 2025-11-01

**Soundness:** 3
**Presentation:** 3
**Contribution:** 3
**Rating:** 8
**Confidence:** 4

**Summary:**

In this paper, an algorithm called InfoTok is proposed for adaptive video tokenization. The proposed algorithm determines the token length based on ELBO for near optimal compression. For compression, InfoTok utilizes the ELBO based token selection. Thus, it can be integrated on the top of existing tokenizer architectures. Also, the proofs for the mathematical theorems are provided. The experimental results show that the proposed algorithm outperforms the existing methods.

**Strengths:**

- This paper clearly describes the proposed algorithm and is easy to follow.
- This paper provides the proofs for the theorems.
- The proposed algorithm achieves good performance on various benchmark tests.

**Weaknesses:**

- It would be helpful if the paper included a discussion of the limitations of the proposed approach.
- Since the method seems general enough to be applied to images, it would be helpful to explain the rationale for limiting the experiments to videos.
- typo
  - L184: an more accurate -> a more accurate

**Questions:**

Though I have carefully reviewed the paper including the appendix, I did not observe any major drawbacks and I believe the proposed algorithm makes a meaningful contribution to adaptive video tokenization. Please see my minor concerns in the weakness section.

---

> ### Author Response · Authors · 2025-11-16
>
> We sincerely thank the reviewer for the appreciation of our contribution and the careful review!
> - [Discussion of limitations] We are delighted to discuss more on the limitations, and we have already included in the updated version. We attach the paragraph here for your reference.
> > While INFOTOK demonstrates significant gains in compression efficiency for video reconstruction, this study has a few limitations. First, our principled ELBO-based router introduces a modest computational overhead by requiring one additional decoder pass to estimate the video's information complexity. Although this is significantly more efficient than the multi-pass search required by prior adaptive methods, further research could explore lighter-weight router mechanisms, such as estimating complexity directly from the encoder's latent representations, to eliminate this extra step entirely. Second, our evaluation is primarily focused on reconstruction fidelity (e.g., PSNR, FVD) as a proxy for representational quality. As noted, we did not extend our experiments to downstream applications such as video generation or action understanding, which was beyond our current scope due to the substantial computational resources required. Investigating how INFOTOK's adaptive token sequences impact the performance and efficiency of large-scale generative or video-understanding models remains a key direction for future work.
> - [Experiments on videos only] Your understanding is absolutely correct that our method is general enough to be applied to images as well. As we have discussed in the footnote in page 3, We do not include image experiments since the amount of tokens used for image representation is relatively small, and the incentive to optimize it is less sufficient. In addition, one of the reason that InfoTok is beneficial to video tokenization is that it can effectively reduce the temporary redundancy in videos whose subsequent frames have higher log-probability given the first few frames. This temporal issue does not exist in images, and so we expect InfoTok would work better for video tokenization tasks.
>
> Again, we thank the reviewer for the recognition of our work, and please let us know if there is any additional question we can help address.

---

### Author Response · Authors · 2025-11-25

With the discussion period having run for approximately two weeks, we want to thank Reviewer t2hE for raising their score to 8 and Reviewer 66nG for confirming their satisfaction with our response. We kindly invite Reviewer 5w2J to let us know if we have resolved their concerns. We are happy to provide further clarifications if needed. Thanks again for your time, thoughtful feedback, and engagement with our work!

---

### Author Response · Authors · 2025-12-02
**Comment for new AC**

Dear Area Chair,

We understand you have been assigned to our submission on short notice. To assist you in your assessment, we would like to briefly summarize the paper’s status and the main improvements made during the rebuttal phase.

**Review Status & Context**

The paper originally received very positive reviews (ratings: 8, 6, 8), where all reviewers agreed on the technical novelty, theoretical contribution, and experimental superiority. **As confirmed by our general comment on Nov 24 (before the incident)**, during the first week of the rebuttal, we were delighted that reviewer t2hE increased the score from 6 to 8, and reviewer 66nG acknowledged we addressed the concerns and may increase it to 10 pending resolution of other authors' concerns (and we have addressed).

**Key Rebuttal Improvements**

We have uploaded a revised PDF and detailed responses addressing the primary feedback:
- Add Contents: Added a section discussing computational overhead. Added a paragraph on extending InfoTok's principles to other domains like audio and 3D data, and confirmed scope focus on video due to higher token optimization incentive and temporal redundancy.
- ELBO/Architecture Details: Clarified that the adaptive compressor is general and the VAE's ELBO conveniently serves as the router.
- Visualization Added: Included a new figure (Figure 3) showing reconstruction fidelity at different token rates and uploaded external video visualizations.

We have addressed all concerns effectively well, and we believe that the work could be a strong case as an oral/spotlight presentation due to its clear and strong theoretical justifiaction, as well as effective empirical improvements.

Thank you again for your time and for taking on this assignment under challenging conditions.

---

### Meta-Review · Area_Chair_Nn5m · 2026-01-04

**Summary:**

This paper studies adaptive discrete video tokenization and proposes InfoTok, a method that allocates token budgets according to the information content of the input using an ELBO-based criterion. A key strength of the work is that it provides a clear theoretical justification for adaptive token length, and translates this insight into a practical tokenizer that can be integrated with existing architectures. The reviewers’ main questions concerned how broadly the approach applies beyond videos, whether the theoretical formulation and notation were sufficiently clear, and whether the empirical results were supported by convincing qualitative evidence. The authors have addressed most of the reviewers’ concerns during the rebuttal phase, improving the paper’s clarity and overall presentation. Taking these responses into account, I find the paper technically solid and recommend accepting this paper.

**Reviewer Concerns:**

The rebuttal addressed the substantive reviewer concerns, including questions about generality, clarity of the theoretical sections, and qualitative evaluation. The remaining limitation is the focus on reconstruction-based evaluation rather than downstream tasks, which is acknowledged by the authors and does not materially weaken the contribution.

**Reviewer Scores:**

Reviewer 5w2J would likely keep their score unchanged, as only minor issues were raised and resolved. Reviewer t2hE would likely have increased their score given that their technical questions were addressed. Reviewer 66nG indicated that the rebuttal resolved their concerns and would likely have maintained or increased their score.

---

### Decision · Program_Chairs · 2026-01-26

Accept (Oral)